# LptM promotes oxidative maturation of the lipopolysaccharide translocon by substrate binding mimicry

Yiying Yang [1], Haoxiang Chen[1,7], Robin A. Corey [2,6,7], Violette Morales[1,7], Yves Quentin [1], Carine Froment[3,4], Anne Caumont-Sarcos[1], Cécile Albenne[1], Odile Burlet-Schiltz[3,4], David Ranava [1], Phillip J. Stansfeld [5], Julien Marcoux [3,4] & Raffaele Ieva [1] ✉

Insertion of lipopolysaccharide (LPS) into the bacterial outer membrane (OM) is mediated by a druggable OM translocon consisting of a β-barrel membrane protein, LptD, and a lipoprotein, LptE. The β-barrel assembly machinery (BAM) assembles LptD together with LptE at the OM. In the enterobacterium *Escherichia coli*, formation of two native disulfide bonds in LptD controls translocon activation. Here we report the discovery of LptM (formerly YifL), a lipoprotein conserved in *Enterobacteriaceae*, that assembles together with LptD and LptE at the BAM complex. LptM stabilizes a conformation of LptD that can efficiently acquire native disulfide bonds, whereas its inactivation makes disulfide bond isomerization by DsbC become essential for viability. Our structural prediction and biochemical analyses indicate that LptM binds to sites in both LptD and LptE that are proposed to coordinate LPS insertion into the OM. These results suggest that, by mimicking LPS binding, LptM facilitates oxidative maturation of LptD, thereby activating the LPS translocon.

Gram-negative bacteria surround and protect their cytoplasm with a multilayered envelope formed by an inner membrane (IM), an outer membrane (OM) and a thin peptidoglycan layer, sandwiched in between. Both lipid bilayers carry out crucial functions, including nutrient uptake and energy metabolism, as well as cytoplasm detoxification and protection against noxious chemicals, thereby promoting adaptability to a wide range of niches[1]. The OM forms the foremost barrier to cellular access and therefore, in the context of bacterial multidrug resistance, the OM is a particularly attractive drug target for the development of new antimicrobials[2,3].

Lipopolysaccharide (LPS) is a major structural component that accumulates in the external leaflet of the OM lipid bilayer. LPS typically contains six saturated acyl chains, surface displayed carbohydrates and negatively charged phosphate groups that are stabilized by divalent cations. This therefore structures a densely-packed layer that shields the cell from lipophilic molecules, detergents and antimicrobial compounds. LPS is also a modulator of protein homeostasis and peptidoglycan remodeling, regulating the overall architecture of the bacterial envelope[4–7]. In addition to its protective function, LPS acts as an endotoxin with potent immunogenic activity and represents a major target of the innate immune response[8].

Anterograde LPS transfer across the bacterial envelope is mediated by a dedicated LPS transport (Lpt) pathway. In the enterobacterial model organism *Escherichia coli*, the Lpt pathway relies on the activity

[1]Laboratoire de Microbiologie et Génétique Moléculaires (LMGM), Centre de Biologie Intégrative (CBI), Université de Toulouse, CNRS, Université Toulouse III - Paul Sabatier (UT3), Toulouse 31062, France. [2]Department of Biochemistry, University of Oxford, Oxford OX1 3QU, UK. [3]Institut de Pharmacologie et de Biologie Structurale (IPBS), Université de Toulouse, CNRS, Université Toulouse III - Paul Sabatier (UT3), Toulouse 31077, France. [4]Infrastructure Nationale de Protéomique, ProFI, FR 2048, Toulouse, France. [5]School of Life Sciences and Department of Chemistry, Gibbet Hill Campus, The University of Warwick, Coventry CV4 7AL, UK. [6]Present address: School of Physiology, Pharmacology and Neuroscience, Biomedical Sciences Building, Bristol BS8 1TD, UK. [7]These authors contributed equally: Haoxiang Chen, Robin A. Corey, and Violette Morales. ✉e-mail: raffaele.ieva@univ-tlse3.fr

of 7 proteins, known as LptA-G[9,10]. Impairment of the Lpt pathway causes accumulation of LPS in the IM[6], where it undergoes subsequent modification by colanic acid[11], a polysaccharide produced upon envelope damage[12]. The terminal link of the Lpt pathway consists of an OM translocon that inserts LPS into the external leaflet of the OM. The OM LPS translocon, composed by an integral OM protein, LptD, and a cognate OM-associated lipoprotein, LptE, acquires a peculiar "plug-and-barrel" architecture[13–15]. LptD consists of an amino (N)-terminal periplasmic β-taco domain and a carboxy (C)-terminal 26-stranded β-barrel transmembrane domain. The β-taco domain receives LPS from a structurally analogous periplasmic protein, LptA. The C-terminal β-barrel domain defines a large internal lumen that is partly plugged by LptE[14–16]. After docking onto the LptD β-taco domain, LPS enters the OM via a mechanism involving a membrane-embedded interface between the β-taco and the β-barrel domains of LptD. This region is proximal to the β-barrel lateral gate that forms between the first and the last β-strands of the LptD transmembrane domain[15–19].

Biogenesis of the OM LPS translocon requires the cooperation of the β-barrel assembly machinery (BAM) and the periplasmic disulfide bond formation machinery (Dsb). BAM is a heteropentamer (BamABCDE) that folds integral OM proteins, such as LptD, into membrane-spanning β-barrel structures[20,21]. The catalytic, integral OM protein BamA and the lipoprotein BamD are two essential BAM subunits that directly coordinate LptD and LptE assembly to form a plug-and-barrel structure[22]. Chaperones and proteases help preventing the accumulation of off-pathway folding intermediates that would hamper OMP biogenesis if blocked at the BAM complex[23,24]. Within LptD, two native disulfide bonds between pairs of non-consecutive cysteine residues interlink the β-taco and the β-barrel domains, thereby activating the LPS translocon[25]. This complex LptD oxidation pattern occurs via a stepwise maturation process. The first two consecutive cysteines in the β-taco domain are initially oxidized by the oxidase DsbA, generating a non-native disulfide bond. After engaging with the BAM complex for assembly of LptD together with LptE in the OM, acquisition of native disulfide bonds in LptD activates the translocon[26]. Typically, DsbC promotes disulfide bond shuffling in secretory proteins[27], however the role of this enzyme in LptD oxidative folding has been debated[25,28].

The development of Lpt-targeted drugs requires detailed understanding of the process of LPS transport to the OM. Here we report the discovery of LptM (formerly YifL) as a component of the OM LPS translocon. LptM binds the membrane-embedded portion of the translocon, making contacts with sites in both LptD and LptE that are proposed to mediate LPS insertion into the OM. Thus, LptM stabilizes an active conformation of the OM LPS translocon promoting its assembly by the BAM complex and oxidative maturation. LptM is a component of the Lpt pathway that controls activation of the OM LPS translocon by mimicking substrate binding.

## Results
### The OM LPS translocon stably interacts with LptM
We discovered that the *E. coli* OM LPS translocon, LptDE, interacts with an uncharacterized lipoprotein, formerly known as YifL, that we have renamed LptM (LptD oxidative maturation-associated lipoprotein). We identified LptM upon expression and Ni-affinity purification of the LptDE[His] complex (harboring a C-terminally poly-histidine tagged variant of LptE) using the mild, non-ionic detergent n-dodecyl-β-D-maltopyranoside (DDM). The elution fraction was analyzed by sodium dodecyl sulfate (SDS)-polyacrylamide gel electrophoresis (PAGE) followed by Coomassie brilliant blue staining, revealing efficient co-purification of LptD along with the bait protein LptE[His] (Fig. 1a, lane 1). Analysis of the same elution fraction by blue native (BN)-PAGE showed that the OM LPS translocon migrates with an apparent molecular weight of ~140 kDa (Fig. 1a, lane 3). To verify the protein content of this major gel band, we employed trypsin digestion and MALDI-TOF/TOF

tandem mass spectrometry (MS). Surprisingly, in addition to LptD and LptE, this analysis identified LptM, an uncharacterized lipoprotein of ~6 kDa (Supplementary Fig. 1a–c). To explore the role of this factor, we purified the LptDE complex from a strain lacking the *yifL/lptM* gene. Although the yields of LptE[His] and LptD resolved by SDS-PAGE were unchanged (Fig. 1a, lanes 1 and 2), the intensity of the blue native 140 kDa band was reduced (Fig. 1a, lanes 3 and 4). Part of purified LptE[His] and LptD originated a slightly slower-migrating BN gel band that was most prominent in the sample lacking LptM. The accumulation of a slower-migrating form of the LptDE complex could be indicative of an effect of LptM on the native organization of the OM LPS translocon.

To address whether LptM can stably interact with the OM LPS translocon, a C-terminally poly-histidine tagged version of *E. coli* LptM (LptM[His]) was overproduced together with LptD and LptE. Ni-affinity chromatography revealed efficient purification of the OM LPS translocon subunits along with the bait protein (Fig. 1b). Most notably, native MS demonstrated that LptD, LptE and LptM[His] formed a complex which is larger by 5900 Da ± 25 Da compared to the LptDE[His] heterodimer (Fig. 1c). This mass difference fits well with the theoretical molecular weight of mature tri-acylated LptM, 5888 Da. Together, these results indicate that LptM stably associates with LptDE forming a 1:1:1 heterotrimeric complex.

Next, we asked whether, by associating with LptDE, LptM integrates the transenvelope Lpt pathway, including the periplasmic LPS chaperone LptA that bridges the IM and OM Lpt complexes. An interaction of LptD with LptA can be monitored by site-directed photocrosslinking, introducing the photoactivatable amino acid analog *para*-benzoyl-phenylalanine (*p*Bpa)[29] in place of Y63 at the N-terminal region of the LptD β-taco domain, which interacts with LptA[13]. We observed that the population of LptD that stably associates with LptM[His] could efficiently crosslink LptA in proximity of LptD Y63 (Fig. 1d, lanes 1 and 2). In a control experiment, *p*Bpa was introduced in place of Y347, a position buried in the lumen of the LptD β-barrel, which inefficiently crosslinked LptA (Fig. 1d, lanes 3 and 4). In addition, LptA and, albeit to a lower extent, LptB were co-isolated along with DDM-solubilized LptM[His] (Supplementary Fig. 2a). Of note, the overproduction of LptM[His] had no detectable effect on bacterial growth nor on the OM permeability barrier to the large molecular weight antibiotic vancomycin (Supplementary Fig. 2b), suggesting that its association with the Lpt pathway did not interfere with LPS transport and OM homeostasis. We conclude that LptM stably interacts with the OM LPS translocon LptDE, thus integrating the transenvelope Lpt pathway.

### LptM is a lipoprotein conserved in *Enterobacteriaceae*
The gene *lptM* codes for a small lipoprotein of ~6 kDa. The LptM lipobox cysteine residue is followed by a glycine residue (Fig. 1e), suggesting that the protein is transported to the OM[30], in agreement with our finding that LptM interacts with the OM complex LptDE. LptM includes the annotated Pfam domain, PF13627 (Prokaryotic lipoprotein-attachment site), overlapping with the protein lipobox and a short N-terminal region of the mature protein. This motif was used to search *lptM* putative orthologs in a group of 2927 bacterial genomes (comprising at least one genome per bacterial family of those present in the bacterial Genome Taxonomy Database, GTDB)[31]. *lptM*-like genes were found in ~79% of Gamma- and 57% of Alpha-proteobacteria, and were virtually absent in non-proteobacterial genomes (Supplementary Fig. 3a–d, see also Supplementary Methods). The degree of conservation of chromosomal neighboring genes surrounding *lptM* in Proteobacteria suggests inheritance of this gene from a common ancestor. Thus, the identified *lptM* genes can be considered orthologs. The amino acid sequence length downstream PF13627 is generally short with less than 70 amino acids (Supplementary Fig. 3a). Both the N-terminal PF13627 and the C-terminal moiety of the protein are conserved in a recently redefined *Enterobacteriaceae* family (Fig. 1e

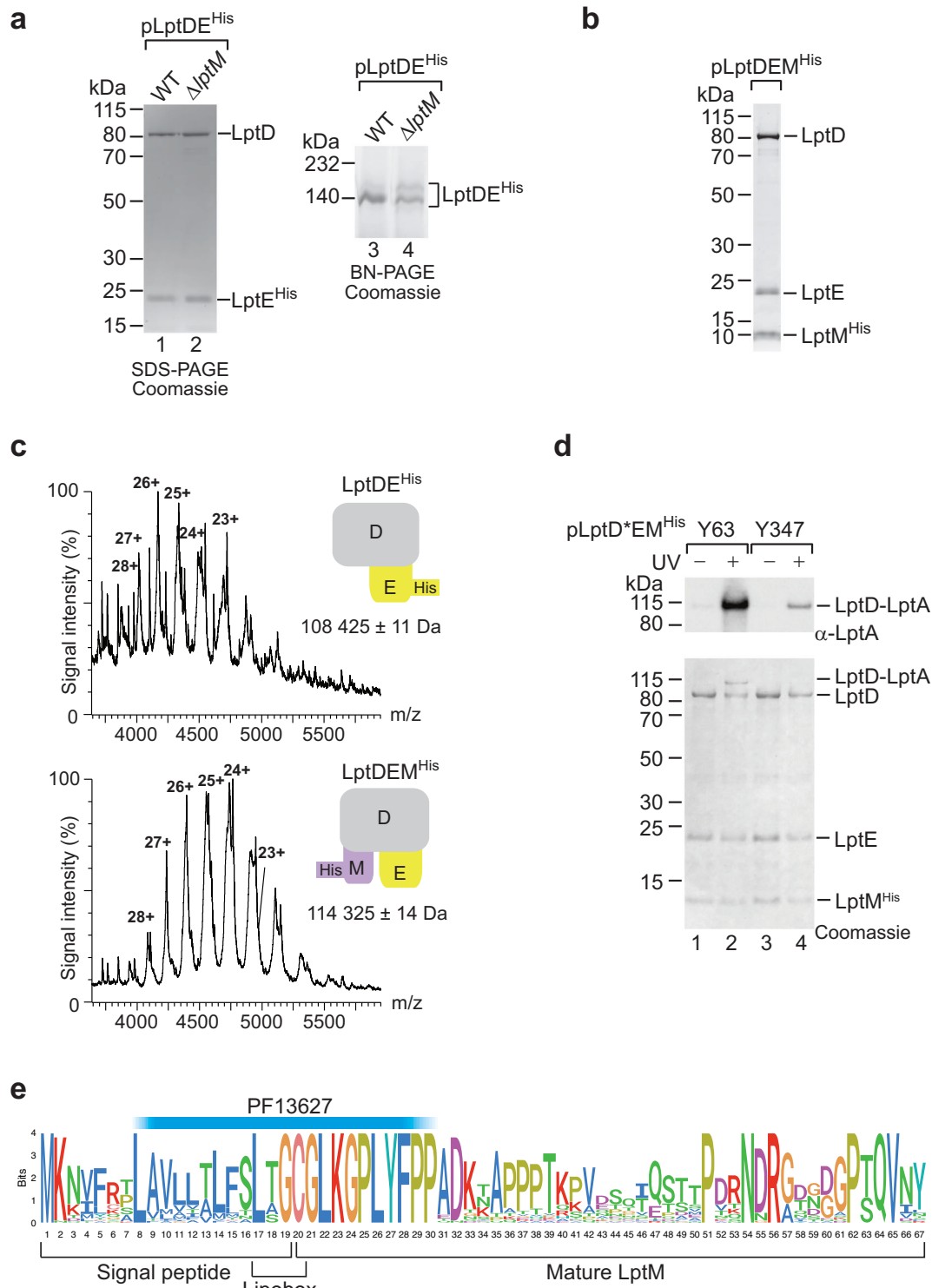

and Supplementary Fig. 4), whereas other Proteobacteria harbor a lipoprotein containing the N-terminal PF13627 motif followed by a C-terminal region that is poorly conserved with that of *E. coli* LptM (Supplementary Fig. 3e; see also Supplementary Methods). Taken together, our taxonomic analysis strongly suggests that distinct sub-regions of LptM from *Enterobacteriaceae* are functionally important.

### LptM promotes efficient assembly of the OM LPS translocon at the BAM complex

The BAM complex plays a key role in the assembly of LptD and LptE as a two-protein plug-and-barrel complex. We noticed that purification of DDM-solubilized LptM[His] co-isolated, to some degree, BAM subunits (Supplementary Fig. 2a, lane 4), raising the hypothesis that LptM could be assembled with the LPS translocon during its biogenesis at the BAM complex. We thus assessed whether LptM would be required for BAM-mediated assembly of LptD together with LptE. To test this hypothesis, first we generated a double deletion strain lacking *lptM* and *bamB*, which encodes a subunit of the BAM complex that is particularly critical for LPS translocon biogenesis[32]. ∆*bamB* is sensitive to detergents, such as SDS, and to large molecular weight antibiotics, such as vancomycin, which are normally excluded from the OM of wild-type cells. To compare the phenotypes of single and double *lptM* and *bamB* deletion

**Fig. 1 | LptM interacts with the LPS translocon. a** The envelope fractions of wild-type (WT, *lptM*⁺) and Δ*lptM* transformed with pLptDE^His cells were solubilized using the mild detergent DDM and subjected to Ni-affinity purification. The purified OM LPS translocon was analyzed by SDS-PAGE (lanes 1 and 2) and BN-PAGE (lanes 3 and 4) followed by Coomassie Brilliant Blue staining. Data are representative of three independent experiments. **b** DDM-solubilized LptM^His purified from Δ*lptM* cells transformed with pLptDEM^His was analyzed by SDS-PAGE and Coomassie Brilliant Blue staining. Data are representative of three independent experiments. **c** Native mass-spectrometry analysis of LptDE^His and of LptDEM^His purified as in (**a**) and (**b**), respectively. A schematic diagram of each complex is shown for each spectrum; LptD in gray, LptE in yellow, LptM in purple. Top: LptDE^His formed a heterodimeric complex of 108,425 ± 11 Da (LptD, theoretical mass of the mature protein = 87,080 Da; LptE^His, theoretical mass of the mature and tri-acylated protein = 21,348 Da). Bottom: LptDEM^His formed a heterotrimeric complex of 114,325 ± 14 Da (LptD, theoretical mass of the mature protein = 87,080 Da; LptE, theoretical mass

of the mature and tri-acylated protein = 20,248 Da; LptM^His, theoretical mass of the mature and tri-acylated protein = 6,988 Da). The mass difference between the two complexes is 5,900 Da ± 25 Da, corresponding to the mass of mature tri-acylated LptM 5888 Da. **d** Δ*lptM* cells transformed with pLptDEM^His derivative (*) plasmids that carry *p*Bpa at distinct positions in LptD were subjected to UV irradiation as indicated prior to DDM-solubilization and Ni-affinity purification of LptM^His. Upon SDS-PAGE of the elution fractions, proteins were revealed by Coomassie Brilliant Blue staining or Western blotting using an antiserum against LptA. Data are representative of three independent experiments. **e** Logoplot representation of the LptM amino acid sequence in *Enterobacteriaceae*. The plot was obtained from the multiple alignment of 37 amino acid sequences of LptM in representative genomes of a restricted Enterobacteriaceae family reported in Supplementary Fig. 4 (see also Supplementary Information). The location of PF13627 is indicated above the logoplot.

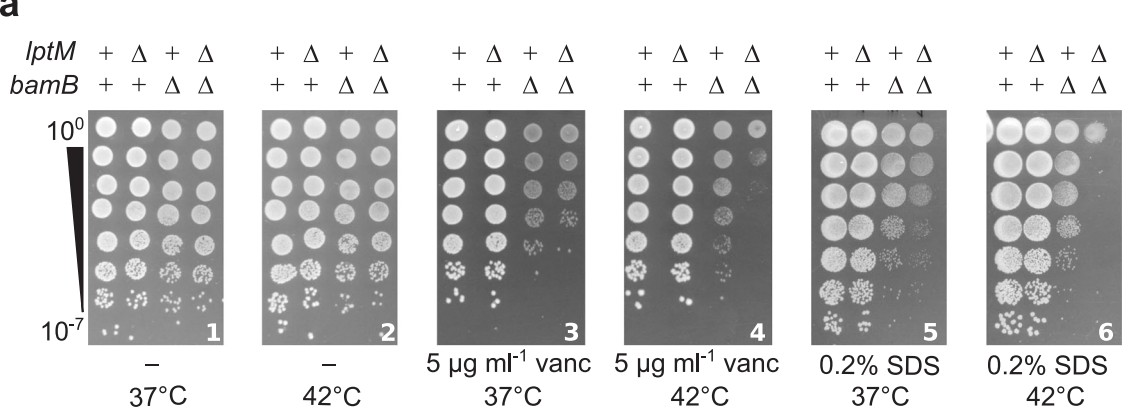

**a**

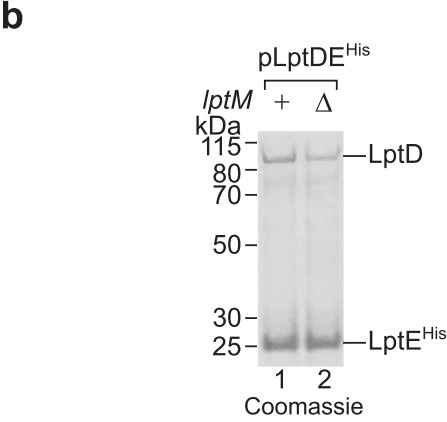

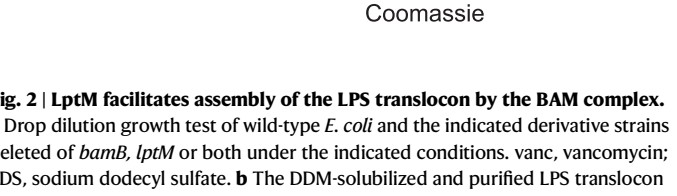

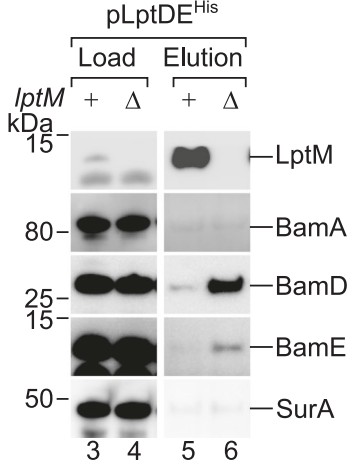

**b**

**Fig. 2 | LptM facilitates assembly of the LPS translocon by the BAM complex. a** Drop dilution growth test of wild-type *E. coli* and the indicated derivative strains deleted of *bamB*, *lptM* or both under the indicated conditions. vanc, vancomycin; SDS, sodium dodecyl sulfate. **b** The DDM-solubilized and purified LPS translocon obtained from *lptM*⁺ or Δ*lptM* cells harboring pLptDE^His was analyzed by Coomassie Brilliant Blue staining to visualize the bait protein (lanes 1 and 2) used for Western blotting with the indicated antisera to identify co-eluted proteins (lanes 3-6). Load: 1.8%; Elution: 100%. Data are representative of three independent experiments.

strains, we used low concentrations of vancomycin or SDS that had only a partial inhibitory effect on the growth of Δ*bamB* and that had no effect on Δ*lptM* (Fig. 2a, frames 3-6). The sensitivity of the double deletion strain to both SDS and vancomycin was considerably accentuated compared to Δ*bamB*, especially at 42 °C (Fig. 2a, frames 4 and 6). This detrimental synergistic effect reveals a genetic association of *lptM* with *bamB*, pointing to a possible role of LptM in LptDE assembly.

Mutations that impair the folding of LptD delay LPS translocon assembly into the OM, thereby prolonging its time of residence at the BAM complex[22,33]. We reasoned that, if LptM was indeed important for efficient assembly of LptDE, the deletion of *lptM* could cause accumulation of the LPS translocon at the BAM complex. To address this hypothesis, LptDE^His was produced either in wild-type (*lptM*⁺) or in Δ*lptM* cells and affinity purified, yielding similar amounts of the bait

protein (Fig. 2b, lanes 1 and 2). Western blot analysis of the elution fractions revealed that LptM was highly enriched in the LptDE$^{His}$ elution fraction obtained from *lptM*$^{+}$ cells (Fig. 2b, lane 5). Strikingly, the BAM subunits BamD and BamE were also detected in the elution fraction obtained from Δ*lptM* cells but not in that obtained from *lptM*$^{+}$ cells, indicating a prolonged interaction of LptDE$^{His}$ with the BAM complex in the absence of LptM (Fig. 2b, lanes 5 and 6). We conclude that LptM is required for efficient release of the OM LPS translocon by the BAM complex.

### Inactivation of *lptM* impairs LptD oxidative maturation

We asked whether LptM influences the oxidative maturation of LptD that occurs concomitantly to LPS translocon assembly by the BAM complex. LptD contains four cysteine residues at position 31, 173, 724 and 725 of its amino acid sequence, hereafter named C1 to C4, respectively (Fig. 3a). Upon transport into the periplasm by the Sec machinery, the two most N-terminal cysteine residues C1 and C2 are oxidized by DsbA, leading to formation of an oxidation intermediate (LptD$^{C1-C2}$). Folding of LptD$^{C1-C2}$ into a β-barrel structure that surrounds LptE[13,22] is followed by the isomerization of the C1-C2 disulfide bond to the C2-C4 bond and further oxidation of the remaining C1 and C3 cysteines[25,26]. Hereafter, we will refer to the mature, fully oxidized form of LptD (LptD$^{C1-C3,C2-C4}$) as LptD$^{Ox}$. At least one disulfide bond between non-consecutive cysteines (C1-C3 or C2-C4) is required for proper LPS translocon function, whereas the earlier assembly intermediate, LptD$^{C1-C2}$, is functionally defective[25,26,28]. The mechanisms mediating LptD disulfide bond formation and isomerization, which are critical for LPS translocon activation, remain only partially understood[25,28,34].

The oxidation states of LptD can be determined by non-reducing SDS-PAGE, as LptD$^{C1-C2}$ migrates slightly faster than reduced LptD (LptD$^{Red}$), whereas LptD$^{Ox}$ migrates slower than both LptD$^{C1-C2}$ and LptD$^{Red}$ (Fig. 3a)[25,26]. Strikingly, our analysis of endogenous LptD in wild-type and Δ*lptM* cells revealed that the level of LptD$^{Ox}$ was reduced in Δ*lptM* and fully restored in a derivative complementation strain harboring plasmid-encoded LptM$^{His}$ (Fig. 3b). Of note, Δ*lptM* cells accumulated some low amount of a faster-migrating form of LptD. To identify this LptD form, we analyzed the migration of LptD produced by strains lacking single components of the periplasmic oxidative folding machinery. The lack of the oxidase DsbA caused the accumulation of the reduced form, LptD$^{Red}$ (Fig. 3c, lane 8)[25], whereas elimination of the thiol-quinone oxidoreductase DsbB and the reductase DsbG had no obvious effect on LptD oxidation under these conditions (Fig. 3c, lanes 9 and 11). In contrast, the lack of DsbC and of its electron donor DsbD caused the accumulation of LptD$^{C1-C2}$ (Fig. 3c, lanes 10 and 12). The LptD migration pattern obtained with the deletion of *lptM* is similar to that obtained with Δ*dsbC* and Δ*dsbD*, suggesting that Δ*lptM* accumulates the LptD$^{C1-C2}$ intermediate (Fig. 3b, lane 2; see also Fig. 3c lane 14). Furthermore, these results indicate that a Δ*lptM* strain fails to efficiently form mature LptD$^{Ox}$. Lack of the periplasmic chaperone/proteases BepA, which can degrade an early off-pathway intermediate of LptD assembly[23,24], did not alter the pattern of LptD bands in Δ*lptM* (Fig. 3c, lanes 13-16), suggesting that BepA does not function synergistically with LptM nor degrades LptD assembly intermediates in Δ*lptM*, under our experimental conditions.

To further analyze the LPS translocon maturation defect caused by the absence of LptM, we overproduced LptD and LptE$^{His}$ either with or without LptM and performed Ni-affinity purifications. As a reference, we also purified translocon mutant forms containing three combinations of Cys-to-Ser pair substitutions in LptD (LptD$^{CCSS}$, LptD$^{CSCS}$ and LptD$^{SCSC}$). With the overproduction of LptM, nickel-affinity purification of LptE$^{His}$ yielded ~95% of LptD in its mature oxidized state, LptD$^{Ox}$ (Fig. 3d, lane 2, and quantifications). In contrast, a significantly lower amount of LptD$^{Ox}$ was obtained in the absence of LptM (Fig. 3d, lane 1). In this sample, approximately one third of LptD was either reduced or only partially oxidized (Fig. 3d, lane 1, and

quantifications). These LptD oxidation intermediate forms included LptD$^{C1-C2}$ (migration similar to that in lane 3 [LptD$^{CCSS}$]) and a LptD form containing a non-consecutive disulfide bond that migrates slower than LptD$^{Ox}$, seemingly corresponding to LptD$^{C2-C4}$ (Fig. 3d, compare lane 1 to lanes 4 [LptD$^{CSCS}$] and 5 [LptD$^{SCSC}$]). We also found that the complex containing LptD$^{CCSS}$ migrates slower on BN gel compared to the complex containing wild-type LptD (Fig. 3e, lanes 1 and 2). These results suggest that, compared to the LPS translocon containing LptD$^{Ox}$, the presence of LptD$^{C1-C2}$ retards translocon migration on BN gels, providing a plausible explanation for the differential BN-PAGE separation patterns of LptDE purified from wild-type cells (containing mostly LptD$^{Ox}$) or from Δ*lptM* cells (containing also partially oxidized LptD forms) (Fig. 1a, lanes 3 and 4, and Fig. 3e, lanes 3 and 4). We conclude that LptM facilitates LptD oxidative maturation, whereas its inactivation causes the accumulation of intermediate oxidation forms, most prominently LptD$^{C1-C2}$ (Fig. 3b, c and e).

### LptM functions synergistically with DsbA during LptD oxidative maturation

The observation that Δ*lptM* accumulates the LptD$^{C1-C2}$ oxidation intermediate suggests that LptM is not strictly required for DsbA activity. Yet LptM could have a different role in LptD oxidative maturation, thus acting synergistically with DsbA. To test this hypothesis, we generated a double deletion strain lacking *lptM* and the cysteine oxidase gene *dsbA* by using a P1 phage transduction protocol. On LB agar plates, the Δ*lptM* Δ*dsbA* strain formed slightly smaller colonies compared to its parental single deletion strains (Fig. 4a, left). Furthermore, the double deletion strain showed an enhanced sensitivity to both vancomycin and SDS (Fig. 4a, center and right), indicative of a pronounced OM permeability barrier defect that was not observed with the single deletion strains. This result implies that, as hypothesized, LptM acts synergistically with DsbA in promoting proper oxidative maturation of LptD.

We also investigated whether the detrimental effect caused by the *lptM dsbA* double deletion was related to impaired LPS transport. LPS extracted from the wild-type and the single deletion strains showed similar SDS-PAGE and silver staining patterns (Fig. 4b, lanes 2-4), whereas slower migrating forms of LPS were obtained from the double *lptM dsbA* deletion strain (Fig. 4b, lane 1). A mass increment of LPS reveals a modification that can be triggered as a consequence of impaired anterograde transport along the Lpt pathway, causing LPS accumulation at the IM and modification with colanic acid[6,11]. Most importantly, the observed LPS alteration in the *lptM dsbA* double deletion strain corroborates our evidence that LptM promotes LPS translocon biogenesis, thereby facilitating efficient LPS transport to the OM.

### In the absence of DsbC, LptM is crucial for cell viability

To gain further insights into how LptM functions together with the oxidative folding machinery, we sought to generate the double deletion strains Δ*lptM dsbC::kan* or Δ*dsbC lptM::kan*. However, we could not combine the deletion of *lptM* with that of *dsbC* by using P1 phage transduction (see Methods for data quantification), suggesting that the double deletion of *lptM* and *dsbC* is synthetically lethal. Indeed, a strain lacking both chromosomal *dsbC* and *lptM*, and encoding DsbC under the control of an arabinose-inducible promoter, P$_{BAD}$, could grow efficiently only in the presence of the inducer (Fig. 4c), indicating that *dsbC* and *lptM* are nearly essential upon inactivation of the other. We conclude that in the absence of LptM, LptD disulfide bond isomerization by DsbC becomes critical.

To gain deeper understanding of the consequences related to impaired LptD disulfide bond isomerization, we challenged cells with the overproduction of LptE and wild-type LptD (LptD$^{CCCC}$) or LptD$^{CCSS}$, *i.e.* a form of LptD that can be converted to LptD$^{C1-C2}$ but that cannot undergo subsequent disulfide bond shuffling as it contains serine

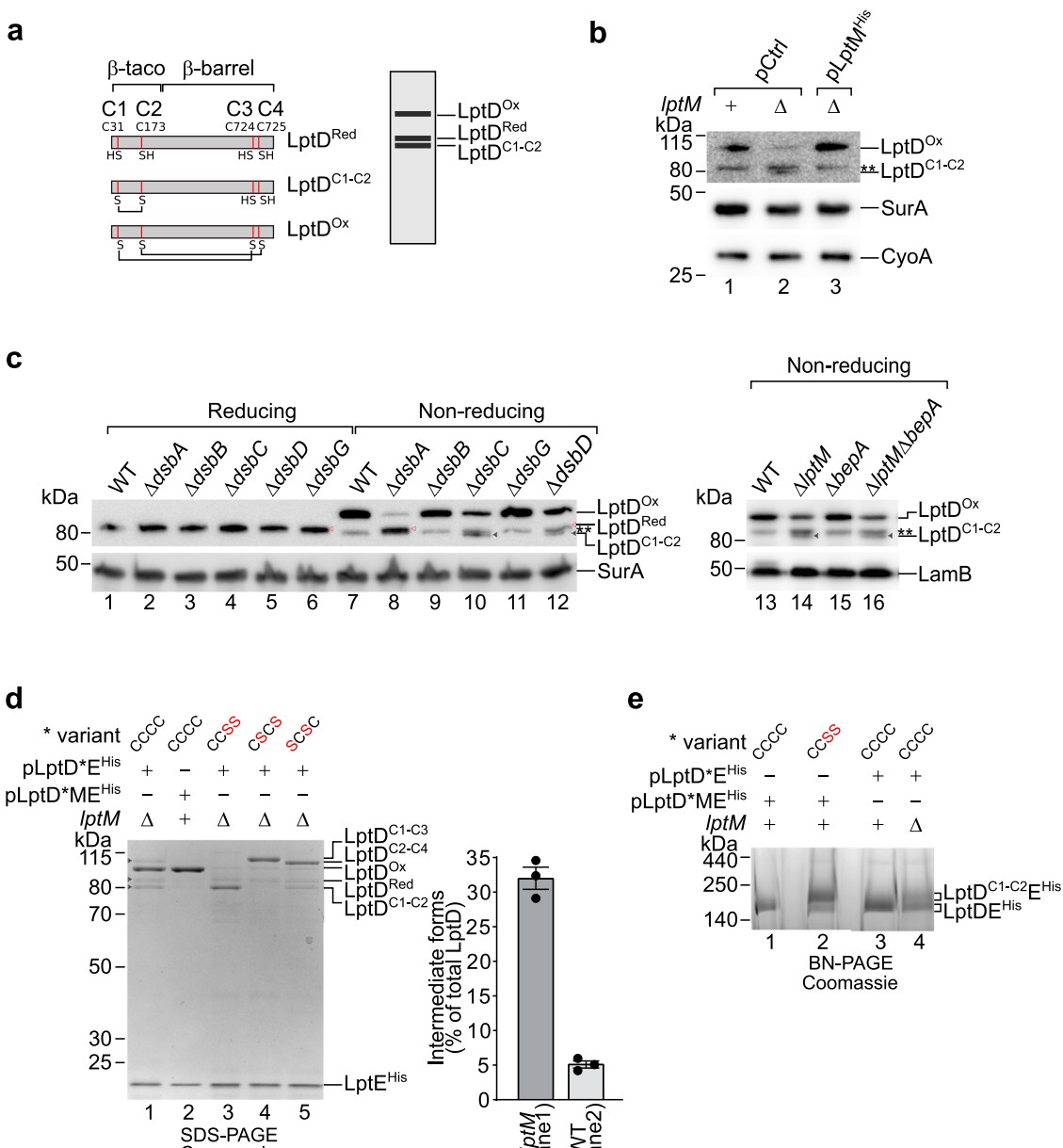

**Fig. 3 | LptM promotes LptD oxidative maturation. a** Schematic representation of mature LptD primary sequence indicating the position and oxidation state of Cys residues in LptD$^{Red}$ (top, left), LptD$^{C1-C2}$ (middle, left) or fully oxidized LptD$^{Ox}$ (bottom, left). A typical migration pattern corresponding to different oxidative states of LptD by non-reducing SDS-PAGE is represented on the right. **b** The total protein contents of wild-type (*lptM*$^+$) and Δ*lptM* strains transformed with an empty vector pCtrl, and the complementation strain Δ*lptM* transformed with pLptM$^{His}$ were separated by non-reducing SDS-PAGE and analyzed by Western blotting using the indicated antisera. **Indicates a non-identified protein band. Data are representative of three independent experiments. **c** The total protein contents of wild-type and the indicated mutant strains were separated by reducing or non-reducing SDS-PAGE and analyzed by Western blotting as in (**b**). Red, empty arrowheads indicate reduced LptD (LptD$^{Red}$), whereas filled, gray arrowheads indicate LptD$^{C1-C2}$. **Indicates a non-identified protein band that migrates slightly faster than LptD$^{Red}$. Data are representative of three independent experiments. **d** Purification of LptE$^{His}$ from

Δ*lptM* cells that overproduced LptDE$^{His}$ (lane 1) or from wild-type cells that overproduced LptDME$^{His}$ (lane 2) or Δ*lptM* cells that overproduced the indicated LptD mutant variant of LptDE$^{His}$ (3-5). *Refers to the LptD Cys-to-Ser variant: wild-type LptD$^{CCCC}$, LptD$^{CCSS}$, LptD$^{CSCS}$ or LptD$^{SCSC}$, as specified on the top of each gel lane. Data are representative of three independent experiments. The signals of any LptD forms in lanes 1 and 2 were quantified. The amount of intermediate forms (LptD$^{Red}$ + LptD$^{C1-C2}$ + LptD$^{C2-C4}$) was normalized to the overall amount of all LptD forms (LptD$^{Red}$ + LptD$^{C1-C2}$ + LptD$^{C2-C4}$ + LptD$^{ox}$). Data are represented as means ± s.e.m. (*n* = 3 independent experiments). Source data are provided as Source Data file. **e** LptDE$^{His}$ containing either wild-type (LptD$^{CCCC}$) or LptD$^{CCSS}$ co-overproduced together with LptM or alone in wild-type or Δ*lptM* cells were Ni-affinity purified and resolved by BN-PAGE, prior to Coomassie Brilliant Blue staining. *Indicates the plasmid-encoded LptD variant, wild-type LptD$^{CCCC}$, LptD$^{CCSS}$, as specified on the top of each gel lane. Data are representative of three independent experiments.

residues in place of C3 and C4. Notably, the overproduction of LptD$^{CCSS}$ impaired growth of Δ*lptM* but not of cells overproducing also LptM (Fig. 4d), implying that the accumulation of LptD$^{C1-C2}$ in the absence of LptM is detrimental. Given the prolonged residence of LptD and LptE at the BAM complex in Δ*lptM* cells (Fig. 2b) and that LptD$^{C1-C2}$ is the

LptD oxidation intermediate that engages with the BAM complex[22,26], it appears reasonable to infer that the detrimental effect caused by LptD$^{CCSS}$ overproduction in the absence of LptM is related to the accumulation of an off-pathway intermediate at the BAM complex, which may interfere with the correct functioning of this essential

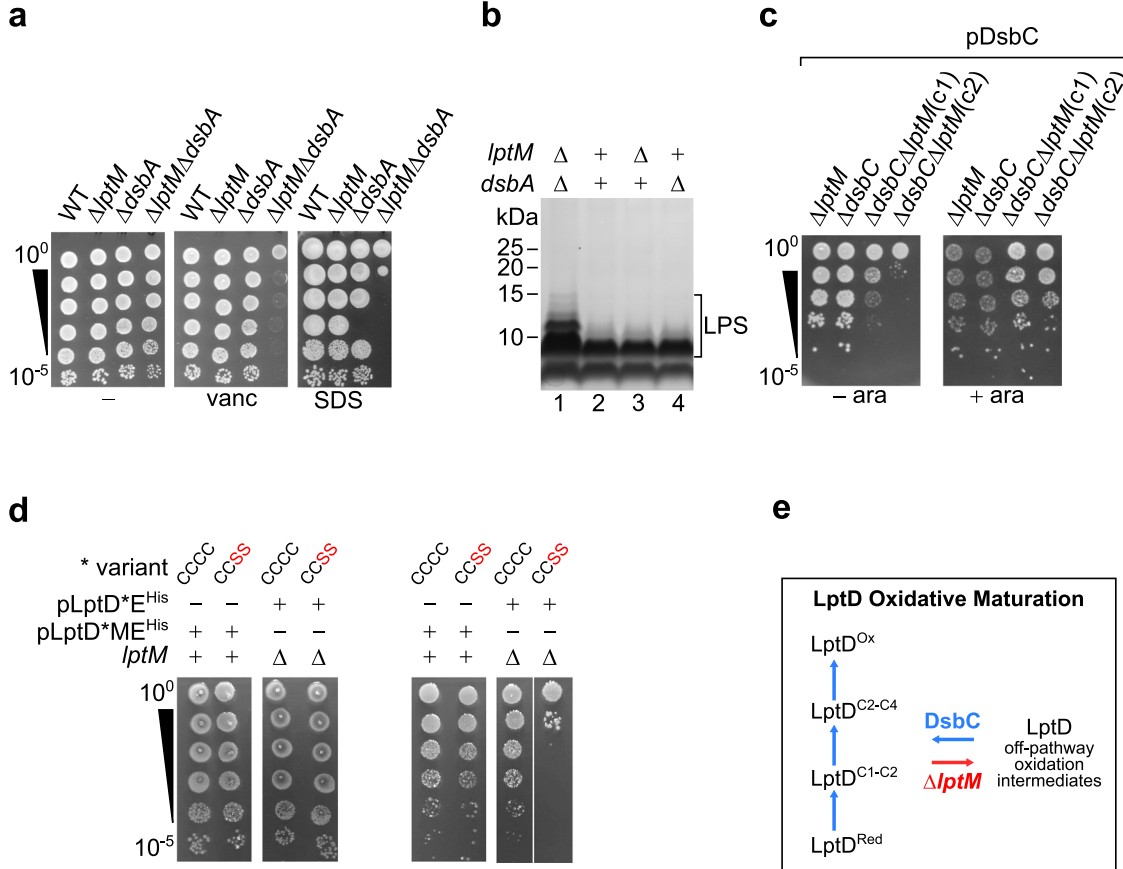

**Fig. 4 | LptM functions synergistically with the oxidative folding machinery.**
**a** Drop dilution growth test of *E. coli* wild-type and the indicated derivative strains lacking *lptM*, *dsbA* or both under different conditions as indicated. vanc, vancomycin; SDS, sodium dodecyl sulfate. **b** LPS was extracted from the indicated strains, resolved by SDS-PAGE and silver stained. Results are representative of four independent experiments. **c** Drop dilution growth test of the indicated derivative strains lacking *lptM*, *dsbC* or both transformed with pDsbC on media lacking or supplemented with 0.02% arabinose (ara). **d** Drop dilution growth test of *lptM*[+] or Δ*lptM* strains transformed with the indicated plasmids pLptD*E[His] or pLptD*ME[His] on media lacking or supplemented with 100 μM IPTG. * indicates variants of plasmid-encoded LptD, the wild-type LptD[CCCC] or the LptD[CCSS] variant, as specified on the top

of each dilution test. **e** Schematic representation of the LptD oxidative maturation pathway. In wild-type cells the four cysteines of LptD are oxidized stepwise: LptD[Red] is first oxidized to form LptD[C1-C2]. Disulfide bond shuffling in the latter generates LptD[C2-C4]. A final event of oxidation forms a disulfide between C1 and C3, thus generating LptD[Ox]. LptM acts synergistically with the disulfide bond formation machinery in facilitating proper oxidation of LptD. In the absence of LptM (Δ*lptM*, red arrow), a considerable fraction of LptD accumulates as off-pathway oxidation intermediates, most prominently LptD[C1-C2]. In this mutant, the assembly of LptD together with LptE stalls at the BAM complex and DsbC becomes essential for cell viability, suggesting that disulfide bond isomerization helps to rescue LPS trans-locon assembly.

machinery. Taken together, our results suggest that, in Δ*lptM*, DsbC helps rescuing LptD off-pathway oxidation intermediates that fail to assemble correctly at the BAM complex (Fig. 4e).

## LptM interacts with the OM embedded portion of the LPS translocon

The structures of LptDE from different organisms have been determined[14–16,18]. To investigate the structural basis of how LptM interacts with LptDE, we used AlphaFold2-multimer to build a predicted model of the complex[35–37]. The top-ranking structures reveal a considerable amount of contact between LptM and LptDE (Fig. 5a and Supplementary Fig. 5), with the N-terminal cysteine of LptM consistently placed in close proximity of the flexible hinge separating the periplasmic and membrane domains of LptD[16,38] that is adjacent to the LptD β-barrel lateral gate (Fig. 5a).

We next investigated how LptM might influence the conformational dynamics of the OM LPS translocon using atomistic molecular dynamics (MD) simulations of our AlphaFold2-multimer model. The presence of the LptM N-terminus close to the LptD hinge was compatible with cysteine pairs C1/C3 and C2/C4 (all reduced) being in disulfide bond distances over a simulation time of 500 ns, whereas C1

and C2 remained more distant from each other (Supplementary Fig. 6a and Supplementary Data 1). Furthermore, the positioning of the N-terminal LptM cysteine means that, when tri-acylated, the acyl tails are positioned at the interface of the LptD β-taco and β-barrel domains (Fig. 5b). This region presents several hydrophobic residues and was proposed to form a membrane-facing hole through which the acyl chains of LPS would get inserted into the OM[19,39]. From our simulations, the presence of LptM at this position breaks several connections of the β-taco with the β-barrel domain (Fig. 5b), which has an impact on the angle of orientation of the LptD β-taco domain (Supplementary Fig. 6b) and causes destabilization of the N-terminal loop of LptD that precedes the β-taco domain (Supplementary Fig. 6c). LptM also affects the LptD β-barrel lateral gate, reducing the number of H-bonds formed between β-strands 1 and 26 at their periplasm-facing side (Fig. 5c). This is in line with a previously proposed model of LptD lateral gate opening[18] and suggests that LptM may reduce the energetic barrier necessary for β-strand unzipping at the gate[15,38].

Aside from these interactions, our model also predicts that LptM inserts into the lumen of the LptD β-barrel, establishing several interaction points with both LptD and LptE. These include a salt bridge formed between LptM K23 and LptD E275 (Supplementary Fig. 6d),

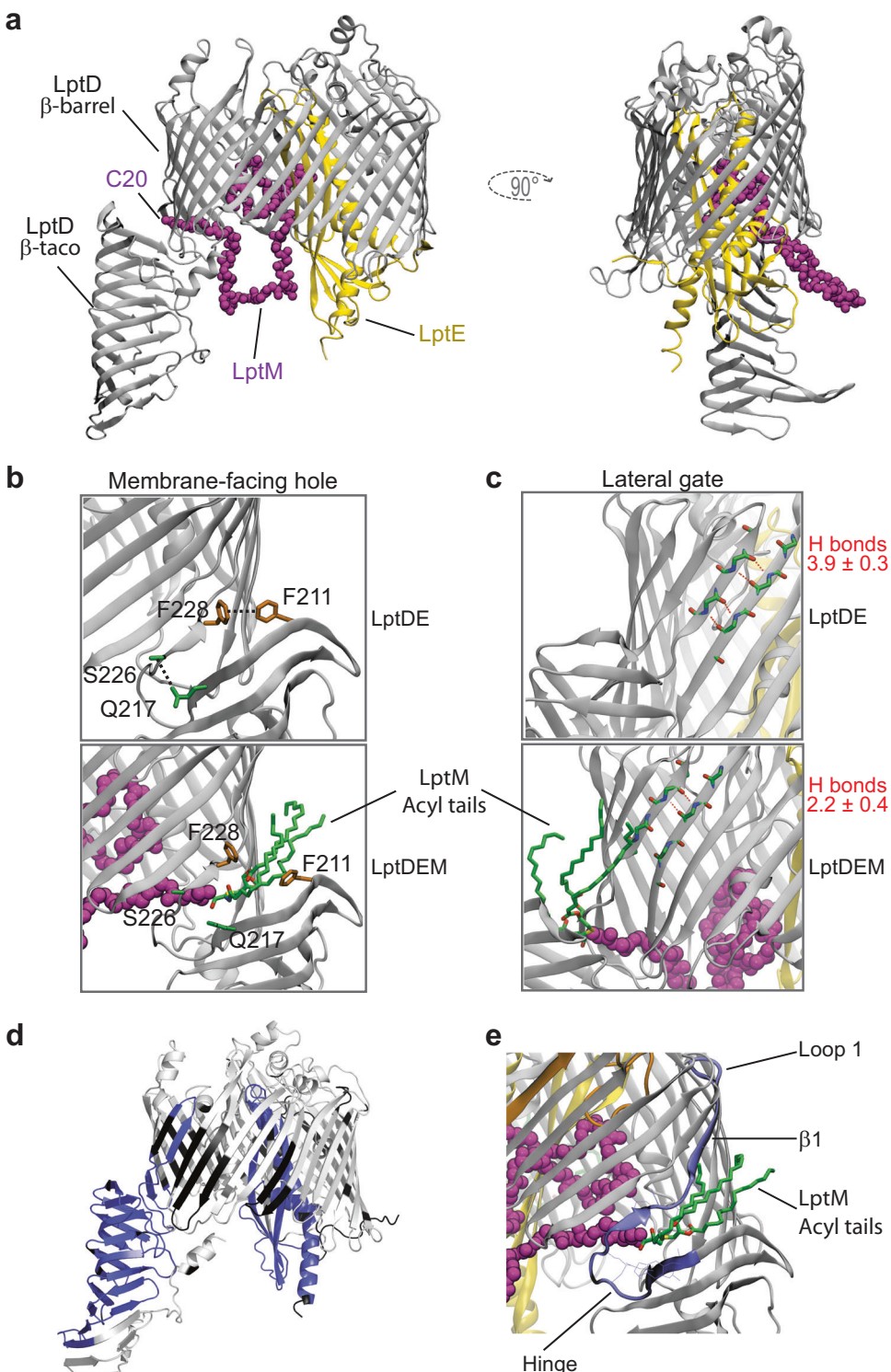

**Fig. 5 | LptM binds the membrane-embedded portion of the LPS translocon, influencing LPS-interaction sites. a** View of top ranking AlphaFold2 model of LptDE-LptM. LptD is shown as gray cartoon, LptE as yellow cartoon, and the LptM backbone is shown as purple spheres. **b** Zoom in on the LptD hinge between the β-taco and β-barrel in either the LptDE (top) or LptDE-LptM (bottom) systems. Image from a snapshot following 500 ns of MD simulation. Several interactions are made between the β-taco and β-barrel in LptDE but broken when LptM is present (purple spheres and green sticks), including between the residues shown. **c** View of the LptD lateral gate with quantification of hydrogen (H)-bond number from 3 × 500 ns simulations for the LptDE and LptDE-LptM systems using gmx hbond.

Representative H-bonds are shown as dashed red lines, as computed using VMD. **d** Differential deuterium uptake between the LptDE translocon in the presence or absence of LptM after statistical analysis with Deuteros, showing in blue regions that are significantly protected upon LptM binding. **e** View of the AlphaFold2 model plus modeled LptM tri-acylation, showing the position of the LptM N-terminus and tri-acylation (purple spheres and green sticks, respectively) in relation to the LptD 212-240 peptide, which is protected from deuteration by LptM in the HDX-MS experiments. Residues from the 212-240 peptide which are in contact with LptM (<0.3 nm) are shown as lines.

which are highly conserved in Enterobacteriaceae (Supplementary Fig. 7, left panel). In addition, the hook-shaped C-terminal region of LptM makes contact to LptE, as well as the inward folded LptD loop 4 that is critical for efficient LptD assembly at the BAM complex[22,26,33]. It should be noted, however, that the AlphaFold2 prediction confidence score decreases specifically for the C-terminal moiety of LptM, making less certain its localization in the ternary LptDEM complex (Supplementary Fig. 5C). In addition, our MD simulations predicted that the LptM C-terminal region is conformationally dynamic (Supplementary Fig. 6c), suggesting that it may coordinate multiple interactions.

Our structural model highlights that LptM mostly interacts with the OM-embedded portion of the LPS translocon. Several experiments corroborated this prediction. First, we purified LptM[His] in cells that express, in addition to endogenous wild-type LptD, also a truncated form of LptD lacking the N-terminal β-taco domain, which can still interact with LptE[14–16]. In line with our AlphaFold2-multimer model, we found that the interaction of LptM with the LPS translocon does not require the LptD β-taco domain (Supplementary Fig. 8a, lane 2). Next, we employed site-directed photocrosslinking to probe the environment surrounding LptM in the attempt to map its interactions with the OM LPS translocon. To this end, pBpa was introduced at 6 different positions of the LptM[His] sequence. Upon affinity purification and western-blotting with LptM-specific antibodies, a number of UV-induced crosslink adducts were detected (Supplementary Fig. 8b). The incorporation of pBpa at position L22 and, to a lower extent, Y27 led to the formation of a UV-dependent adduct of ~100 kDa that contained LptD (Supplementary Fig. 8b, c), which is in line with the AlphaFold2-multimer prediction that the conserved LptM N-terminus interacts with LptD. Two major crosslink adducts of an apparent molecular weight of ~50 kDa were obtained with pBpa introduced in the C-terminal half of LptM (Supplementary Fig. 8b). These adducts corresponded to an interaction of LptM with the abundant OM protein OmpA (~37 kDa) (Supplementary Fig. 8d). As OmpA was not detected upon native purification of LptM[His] (Supplementary Fig. 2a), we speculate that the observed LptM-OmpA crosslink adducts are due to spurious association of the overproduced bait with this abundant OM protein. The inefficient crosslinking of LptM C-terminal moiety with the OM LPS translocon suggests that this segment of LptM may acquire different conformations probably mediating short-lived interactions with LptD and LptE. To further probe for the presence of LptM in the lumen of the LptD β-barrel adjacent to LptE, we introduced pBpa at two amino acid positions in LptE: K70 that faces the periplasm oriented toward LptM and A83 in a β-strand of LptE, which faces the LptD β-barrel lumen occupied by LptM in our structural model. Both LptE positions crosslinked to LptM (Supplementary Fig. 8e) supporting our prediction that LptM resides within the portion of the lumen of the LptD β-barrel that is left unoccupied by LptE (Fig. 5a).

### HDX-MS reveals a stabilization effect of LptM on LptDE conformational dynamics

We also employed hydrogen-deuterium exchange mass spectrometry (HDX-MS) to monitor the dynamics and solvent accessibility of LptD and LptE in the LPS translocon produced in the presence or absence of LptM and purified in detergent micelles. We obtained 93% and 75% sequence coverage for LptD and LptE, respectively, and observed that the deuteration was very slow for the membrane-spanning portion of the LptD β-barrel domain and much faster in its periplasmic region, and extracellular loops (Supplementary Fig. 9), as expected for an integral membrane protein[40,41]. Less expected was the relatively high deuteration rates obtained for LptE, as this protein buries in the internal lumen of the LptD β-barrel. Comparing the deuteration of the LPS translocon with and without LptM, we observed that LptM stabilizes distinct regions of the OM LPS translocon in a statistically significant manner (Fig. 5d and Supplementary Figs. 10–12, and Supplementary Data 2 and 3). These stabilized regions include LptE

(Fig. 5d and Supplementary Figs. 10–12, and Supplementary Data 2 and 3) and a segment of LptD (212-240) corresponding to the hinge followed by β 1 and loop 1 of the β-barrel domain (Fig. 5e and Supplementary Figs. 10–12, and Supplementary Data 2 and 3). The protection of LptE is in line with the multiple contacts between LptM and LptE in the lumen of the LptD β-barrel domain predicted by our AlphaFold2-multimer model. Similarly, protection of the β-barrel N-terminal peptide (LptD 212-240) fits well with the predicted positioning of the LptM N-terminus in proximity of the LptD β-taco/β-barrel hinge and the β-barrel lateral gate. This segment would also be proximal to the acyl tails of the LptM N-terminal cysteine, which might contribute to its stabilization (Fig. 5e). Notably, a recent HDX-MS analysis of Klebsiella pneumoniae (Kp) LptDE showed that, upon LPS binding, deuteration of the N-terminal region of the LptD β-barrel in proximity to the lateral gate is reduced[17]. Thus, the presence of LptM produces a stabilization effect in proximity of the LptD lateral gate that is similar to that caused by LPS binding. Taken together, our structure modeling and HDX-MS results suggest an LPS-mimicking effect by LptM in the membrane embedded portion of the LPS translocon.

Deuteration of the β-taco domain was also decreased in the presence of LptM. A bimodal isotopic distribution for peptides of the β-taco domain was previously reported for Kp LptDE[17]. Based on this result, it was suggested that the β-strands of the β-taco domain undergo concerted closing and opening motions that occur with a kinetic slower than the rate of hydrogen/deuterium exchange (EX1 regime). For peptides of the central and C-terminal region of the β-taco domain, LPS binding favored a mixed kinetics of deuterium uptake, whereas the antimicrobial peptide thanatin, which also binds the LptD β-taco domain[42], shifted the equilibrium toward a slower uptake kinetics[17]. We observed a similar bimodal isotopic distribution for peptides of the β-taco (LptD 70-218) of the LptDE dimer in the absence of ligands, as previously reported[17] (Supplementary Fig. 12). However, at this stage, we cannot exclude the presence of two populations that do not interconvert during the labeling time period, thus generating spectra that would resemble those obtained with EX1 regime kinetics. Strikingly, the presence of LptM abrogated this bimodal regime and shifted the mixed kinetics equilibrium toward the slower uptake behavior (Fig. 5d, and supplementary Figs. 10 and 12), suggesting an effect similar to that shown for Kp LptDE in the presence of thanatin. Taken together, these results provide strong evidence that the extensive contacts predicted between LptM and distinct regions of the LPS translocon may have important regulatory roles.

## Discussion

Here we report the discovery of the lipoprotein LptM as a key factor for the oxidative maturation of the OM LPS translocon. We have shown that LptM stably interacts with the OM-embedded portion of the translocon formed by LptD and LptE, promoting its assembly by the BAM complex. Our functional analysis of LptM reveals a mechanism of activation of the LPS translocon upon assembly by the BAM complex, and clarifies the role of disulfide bond isomerization by DsbC during LptD oxidative maturation. Furthermore, our biochemical and structural analyses indicate that LptM integrates the Lpt transenvelope pathway by associating with LptDE and influences domains of the OM translocon reported to be crucial in coordinating LPS transport.

We provide three major lines of evidence demonstrating the critical role of LptM in the assembly of the OM LPS translocon. First, LptM interacts with the LptD β-barrel domain and LptE. This finding is fully supported by our biochemical assays, native-MS analysis, AlphaFold2 modeling, atomistic MD simulations and HDX-MS kinetics. Second, LptM improves the efficiency of LPS translocon assembly at the BAM complex. Third, LptM is required for proper oxidation of LptD and efficient anterograde transport of LPS. Our AlphaFold2-multimer model and atomistic MD simulations provide important clues as to how LptM promotes LptD maturation. LptM stably associates via its

N-terminal region with the membrane-embedded portion of the LPS translocon, making contacts with both LptD and LptE. Furthermore, although with lower confidence, the LptM C-terminal region is also predicted to interact with the LPS translocon. Notably, the LptD assembly defect caused by the deletion of *lptM* (impaired oxidative maturation and prolonged residence at the BAM complex) is somewhat reminiscent of that reported for the variant encoded by the partial loss-of-function allele *lptD4213*[43,44]. This allele encodes a form of LptD, LptD$^{\Delta D330-D352}$, deleted of the segment comprising part of β 7 and of the inward-folded extracellular loop 4 of the β-barrel domain, which is critical for LPS translocon assembly at the BAM complex and its activation[14–16]. LptD$^{\Delta D330-D352}$ accumulates as a C1-C2 intermediate that resides for a prolonged period of time at the BAM complex[26,33], as also obtained for wild-type LptD upon deletion of *lptM*. This observation is consistent with a model where LptM facilitates the correct folding of LptD promoting its maturation to a fully oxidized form. However, in contrast to the LptD$^{C1-C2}$ intermediate produced by the *lptD4213* allele that is partly degraded by the chaperone/protease BepA[23], the LptD$^{C1-C2}$ intermediate produced in the absence of LptM does not appear to be influenced by this periplasmic factor.

Formation of native disulfide bonds in LptD is carefully controlled and occurs only after the β-barrel domain of LptD engages with the BAM complex to assemble together with LptE into the OM[22,26]. Despite the fact that native disulfide bonds are crucial for LPS translocon activation, DsbA is not essential for viability, indicating that spontaneous oxidation sustains formation of a sufficient amount of properly oxidized LptD[34]. Our results indicate that, with the inactivation of the oxidative folding machinery, LptM is particularly critical for efficient LPS translocon activity. Thus, LptM acts independently of, but synergistically with, the oxidative folding machinery in promoting LPS translocon activation. As indicated by our structural modeling and HDX-MS analysis, LptM binds and stabilizes regions of the OM-embedded portion of the translocon. By doing so, LptM stabilizes a conformation of the translocon that can most effectively be oxidized to form native disulfide bonds, even in the absence of a functional oxidative folding machinery (Fig. 4a and c).

Our results show that LptM is not required to generate the LptD$^{C1-C2}$ oxidation intermediate, as this accumulates in its absence (Fig. 3b and d). Instead, LptM plays an important role after this first LptD oxidation event mediated by DsbA. Different views exist on the mechanisms that mediate disulfide bond isomerization in LptD$^{C1-C2}$. Experimental evidence that DsbC can form an intermolecular disulfide bond with LptD suggested that this enzyme could be involved in the rearrangement of its cysteine oxidation patterns[28]. Yet, the observation that DsbC is not essential for cell viability casted doubts on its role during functional activation of the LPS translocon[25]. Our results clearly indicate that DsbC is crucial for viability in cells that lack LptM. In these cells, LptD and LptE reside at the BAM complex for a prolonged period of time compared to cells that express LptM, which is indicative of a stalled assembly process. Stalling of a LptD assembly intermediate at the BAM complex might have catastrophic effects on the biogenesis of the OM, as it can potentially interfere not only with the maturation of the LPS translocon but also with the ability of the BAM complex to insert other proteins into the OM. Our results imply that DsbC plays a role in rescuing LptD off-pathway intermediates that accumulate in Δ*lptM* (Fig. 4e), thus preventing deleterious consequences on OM biogenesis. In line with this scenario, overproduction of the non-isomerizable mutant form LptD$^{CCSS}$ in Δ*lptM* is highly detrimental, suggestive of a higher propensity of Δ*lptM* to generate LptD$^{C1-C2}$ off-pathway folding intermediates. Taken together, we conclude (i) that LptM acts downstream of the first oxidation event mediated by DsbA and (ii) that, in the absence of LptM, LptD oxidation intermediates that fail to assemble into an active translocon require rescuing by DsbC (Fig. 4e). These considerations support a mechanistic model where LptM stabilizes a conformation of the LPS translocon in which LptD can efficiently complete oxidative maturation.

Finally, our results suggest that LptM, besides being crucial for the correct assembly of the LPS translocon into the OM, can also regulate its function. LptM binds the periplasm-facing luminal region of the plug-and-barrel LPS translocon structure. The N-terminus of LptM interacts with a region of the translocon in proximity of the LptD lateral gate. Intriguingly, our structural model shows that the acyl tails at the N-terminus of LptM would insert into the membrane-facing hole that forms at the interface between the β-taco and the β-barrel domains of LptD reducing the hydrogen-bonding of the lateral gate. This result was validated by our HDX-MS analysis, showing that LptM reduced deuteration of the LptD lateral gate. The stabilization effect of LptM on the lateral gate of LptD is similar to that of LPS-binding observed for *Kp* LptD[17]. The interaction of LptM with LptE, which participate in binding LPS[45], further supports a possible regulatory function. Taken together, our findings show that LptM binds region of the translocon that are crucial in coordinating LPS transport into the OM. This observation suggests that LptM stabilizes an active conformation of the LPS translocon by mimicking its natural substrate. Importantly, LptM also rigidifies the periplasmic β-taco domain of LptD, suggesting tight control of the initial event of LPS docking onto the translocon[17]. We postulate that, during LPS transport, the initial event of LPS binding to the β-taco domain increases translocon dynamics and progressively promotes LPS insertion into the OM via the LptD membrane-facing hole and β-barrel lateral gate. A possible scenario, which will warrant further analyses, is that LPS substitutes LptM in the membrane-embedded portion of the translocon prior to breaching the OM.

The broad conservation of the N-terminal LptM PF13627 protein motif in Proteobacteria and, most notably, the conservation of full-length LptM in *Enterobacteriaceae* suggest that the role of LptM in LPS translocon biogenesis can be generalized to other bacteria. The two essential surface-exposed machineries, the OM LPS translocon and the BAM complex, are emerging as attractive targets for the development of novel antimicrobial compounds[2,3]. In this context, by discovering LptM as a LPS translocon component that is assembled together with LptD and LptE at the BAM complex, our findings provide a key piece of information that can be important for the development of OM-targeting drugs.

## Methods

### Bacterial strains and growth conditions

The *E. coli* strains used in this study are listed in Table 1. All strains are derived from BW25113[46]. Gene deletions were obtained by P1 phage transduction using a P1 phage lysate of the corresponding Keio collection strain[47]. Double gene deletions were obtained by performing a second P1 phage transduction after excising the kanamycin-resistance cassette from the first mutated locus using the heat-curable plasmid pCP20[48]. Transduction of wild-type and Δ*lptM* cells with a Δ*dsbC* P1 phage lysate gave rise to 137 and 0 colonies, respectively, whereas wild-type and Δ*dsbC* cells supplemented with the Δ*lptM* P1 phage lysate gave rise to 570 and 0 colonies, respectively. Thus, to build a *dsbC lptM* double deletion strain, the Δ*lptM* strain was first transformed with pDsbC, harboring the *dsbC* gene under the control of the arabinose-inducible P$_{BAD}$ promoter. This *dsbC* diploid strain was subjected to P1 phage transduction to delete chromosomal *dsbC*. The produced DsbC depletion strain was grown in the presence of 0.02% arabinose.

*E. coli* strains were cultured on M9 minimal medium[49], lysogenic broth (LB) liquid media or LB agar plates, and supplemented with the following antibiotics: 100 μg ml⁻¹ ampicillin, 50 μg ml⁻¹ kanamycin, 30 μg ml⁻¹ chloramphenicol. Serial dilution assays were conducted on LB agar plates supplemented with 5 μg ml⁻¹ or 60 μg ml⁻¹ vancomycin, or 0.2% (w/v) SDS, or 100-400 μM Isopropyl β-D-1-thiogalactopyranoside (IPTG), as indicated.

## Table 1 | Bacterial strains

| Strain name | Reference | Lab Identifier |
|---|---|---|
| BW25113: Δ(araD-araB)567 Δ(rhaD-rhaB)568 ΔlacZ4787(::rrnB-3) hsdR514 rph-1 (wild-type reference) | Grenier et al., 2014[46] | Y1 |
| BW25113 lptM::kan | This study | Y14 |
| BW25113 ΔlptM ompA::kan | Ranava et al., 2021[49] | Y40 |
| BW25113 bamB::kan | Ranava et al., 2021[49] | Y11 |
| BW25113 ΔbamB lptM::kan | This study | Y16 |
| BW25113 dsbA::kan | This study | Y81 |
| BW25113 dsbB::kan | This study | Y82 |
| BW25113 dsbC::kan | This study | Y83 |
| BW25113 dsbD::kan | This study | Y84 |
| BW25113 dsbG::kan | This study | Y85 |
| BW25113 ΔlptM dsbA::kan | This study | Y86 |
| BW25113 ΔdsbC lptM::kan; pDsbC | This study | (P275, P276) |
| BW25113 bepA::kan | This study | Y76 |
| BW25113 ΔlptM bepA::kan | This study | Y78 |

## Table 2 | Plasmids

| Plasmid name | Reference | Lab Identifier |
|---|---|---|
| pCtrl | Ranava et al., 2021[49] | pV3 |
| pLptM$^{His}$ | This study | pYY09 |
| pLptDME$^{His}$ | This study | pYY40 |
| pLptDEM$^{His}$ | This study | pYY43 |
| pLptDE$^{His}$ | This study | pYY68 |
| pDsbC | This study | pVM41 |
| pLptD$^{CCSS}$E$^{His}$ | This study | pVM48 |
| pLptD$^{CCSS}$ME$^{His}$ | This study | pVM49 |
| pLptD$^{CSCS}$E$^{His}$ | This study | pVM50 |
| pLptD$^{SCSC}$E$^{His}$ | This study | pVM69 |
| pLptD$^{\beta-barrel}$EM$^{His}$ | This study | pVM52 |
| pLptM$^{L22Amb-His}$ | This study | pYY23 |
| pLptM$^{Y27Amb-His}$ | This study | pYY24 |
| pLptM$^{V42Amb-His}$ | This study | pYY25 |
| pLptM$^{V50Amb-His}$ | This study | pYY26 |
| pLptM$^{A57Amb-His}$ | This study | pYY27 |
| pLptM$^{Y67Amb-His}$ | This study | pYY28 |
| pLptDME$^{K70Amb-His}$ | This study | pVM73 |
| pLptDME$^{A83Amb-His}$ | This study | pVM71 |
| pLptD$^{Y63Amb}$EM$^{His}$ | This study | pVM86 |
| pLptD$^{Y347Amb}$EM$^{His}$ | This study | pVM87 |
| pEVOL-pBpF | Chin et al., 2002[29] | pEVOL-pBpF |
| pCP20 | Datsenko and Wanner, 2000[48] | pCP20 |
| pBAD33 | Guzman et al., 1995[50] | pBAD33 |

## Plasmid construction

Plasmids and oligonucleotides are listed in Table 2 and Supplementary Table 1, respectively. Plasmid pDsbC was obtained by cloning *dsbC* under an arabinose inducible promoter in the pBAD33 vector[50]. All other plasmids were derived by cloning genes of interest downstream of an IPTG inducible promoter in the pTrc99a vector. pLptM$^{His}$, pLptDE$^{His}$, pLptDME$^{His}$, pLptDEM$^{His}$ and pDsbC were constructed by overlap extension PCR cloning. Site-directed mutagenesis was used to replace specific codons in *lptM*, *lptE* and *lptD* with an amber codon for incorporation of the unnatural amino acid analog *p*-benzoyl-L-phenylalanine (*p*Bpa) by amber suppression[29]. Site-directed mutagenesis was also used to replace cysteine- with serine-encoding codons in *lptD* (pLptDE$^{His}$ or pLptDME$^{His}$). pLptD$^{\beta-barrel}$EM$^{His}$ encoding a version of LptD that lacks amino acids 25-205 (corresponding to the β-taco domain) was constructed by inverse PCR on pLptDEM$^{His}$.

## Taxonomic analysis of LptM

The procedure for the taxonomic analysis of LptM is detailed in the Methods section of the Supplementary Information file. The logoplot of the multiple alignment of LptM amino acid sequences was obtained using ggseqlogo[51].

## Cell fractionation and isolation of protein complexes upon solubilisation with a mild detergent

Whole cell lysates were prepared from cells cultured to mid-exponential phase at 37 °C. Where indicated, cells were supplemented with IPTG for 1.5 h prior to collection. Cells were pelleted by centrifugation, lysed in Laemmli sample buffer (Bio-Rad) and subjected to boiling. Protein affinity purification upon solubilisation of the crude envelope fraction with a mild detergent (native conditions) was conducted as previously described[49]. Briefly, when the cell cultures reached mid-exponential phase (OD600 = 0.5) protein expression was induced by supplementing 200 μM IPTG for 1.5 h prior to cell collection by centrifugation. Cells were resuspended in 20 mM Tris-HCl pH 8 containing an EDTA-free protease inhibitor cocktail (Roche). For the purification of protein samples that had to be analyzed both by reducing and non-reducing SDS-PAGE, the cell resuspension buffer was further supplemented with 50 mM iodoacetamide (Sigma) to alkylate free thiol groups of Cys residues. Resuspended cells were mechanically disrupted using a cell disruptor (Constant Systems LTD) set to 0.82 kPa. The obtained cell lysate was clarified by centrifugation at 6000 x *g*, 4 °C for 15 min. The crude envelope fraction was collected by subjecting the supernatant to ultracentrifugation at 100,000 x *g*, 4 °C for 30 min. To perform affinity purification of protein complexes, the crude envelope fraction was solubilized with 50 mM Tris-HCl pH 8, 150 mM NaCl, 20 mM imidazole supplemented with EDTA-free protease inhibitor (Roche) and 1% (w/v) n-dodecyl-β-D-maltopyranoside (DDM, Merck). After a clarifying spin to remove insoluble material, solubilized proteins were incubated with Protino Ni-NTA resin (Machery-Nagel) for 2 h at 4 °C. After extensive washes of the column with 50 mM Tris-HCl pH 8, 150 mM NaCl, 50 mM imidazole and 0.03% (w/v) DDM, bound proteins were eluted with a similar buffer containing 800 mM imidazole and 10% (w/v) glycerol. Aliquots of the elution fractions were snap-frozen in liquid nitrogen for storage at −80 °C or directly analyzed by SDS or blue native gel electrophoresis.

## MALDI-TOF/TOF mass-spectrometry

Excised bands from BN-PAGE were washed by 25 mM NH$_4$HCO$_3$, pH 7.8-acetonitrile 50:50 (v/v), and digested by trypsin. MS and MS/MS experiments were carried out using α-cyano-4-hydroxycinnamic acid matrix at a concentration of 6 mg ml$^{-1}$ in 50 % (v/v) acetonitrile-0.1% trifluoroacetic acid. Analyses of trypsin digested samples were performed on a MALDI TOF/TOF, in reflector positive mode. Parameters were set to source and grid voltages to 20 and 14 kV, respectively, power laser from 2000 to 3500, extraction delay time, 200 ns; shoot number, 5000. Acquisition range was between 800 and 3500 m/z. Spectra were analyzed using Data Explorer software (Applied Biosystems). MS/MS spectra were acquired using a MS/MS positive acquisition method, with 1 kV positive operating mode, and a CID off mode. The MS/MS spectra were examined and sequenced based on assignment of the N-terminal b-ion and C-terminal y-ion series.

## Native Mass Spectrometry

Prior to native MS analysis, 100 μl LptDE$^{His}$ (7 μM) and LptDEM$^{His}$ (10 μM) samples were desalted in 200 mM ammonium acetate, pH 7.4 supplemented with 0.03% (w/v) DDM using ultrafiltration with MWCO 30 kDa Vivacon 500 (Sartorius) and concentrated to ~10-20 μM. Samples were analyzed on a SYNAPT G2-Si mass spectrometer (Waters, Manchester, UK) running in positive ion mode and coupled to an automated chip-based nano-electrospray source (Triversa Nanomate, Advion Biosciences, Ithaca, NY, USA). The voltages applied to the chip, the sample cone and the ion energy resolving quadrupole were set to 1.8 kV, 200 V and −1.0 V, respectively. Proteins were activated in the collision cell with 200 V trap collision energy and an argon flow of 8 ml min$^{-1}$. The instrument was calibrated with a 2 mg ml$^{-1}$ cesium iodide solution in 50% isopropanol. Raw data were acquired with MassLynx 4.1 (Waters, Manchester, UK) and analyzed manually.

## Automated Hydrogen-Deuterium eXchange coupled to Mass Spectrometry (HDX-MS)

HDX-MS experiments were performed on a Synapt-G2-Si mass spectrometer (Waters Scientific, Manchester, UK) coupled to a Twin HTS PAL dispensing and labeling robot (LEAP Technologies, Carborro, NC, USA) via a NanoAcquity system with HDX technology (Waters, Manchester, UK). Briefly, 5.2 μl of protein at 20 μM were diluted in 98.8 μl of protonated (for peptide mapping) or deuterated buffer (20 mM MES pH 6.5, 200 mM NaCl) and incubated at 20 °C for 0, 0.5, 5, 10 and 30 min. 99 μl of this mixture were then transferred to vials containing 11 μL of pre-cooled quenching solution (500 mM glycine at pH 2.3). A maximally deuterated control was obtained to estimate the amount of deuterium back-change, by incubating 15 μL of LptDE at 50 μM in 285 μL d$_4$-Urea (Sigma) 8 M pH 4.5. After 24 h, 300 μL D$_2$O (Eurisotop) were added to reduce the urea concentration and 5 μL DCl 10% (Eurisotop) were added to reduce pH to 2.5. Deuterium uptakes obtained for this maximally deuterated control are represented in triplicate at t = 30 min in Supplementary Fig. 12, and Supplementary Data 2 and 3. Back-exchange values are shown in Supplementary Fig. 13. After 30 s quench, 105 μL of the mixture were injected to a 100 μL loop. Proteins were digested on-line with a 2.1 mm×30 mm EnzymateTM BEH Pepsin column (Waters Scientific, Manchester, UK). Peptides were desalted for 2 min on a C18 pre-column (Acquity UPLC BEH 1.7 μm, VANGUARD) and separated on a C18 column (Acquity UPLC BEH 1.7 μm, 1.0 mm × 100 mm) by a linear gradient (2% to 40% acetonitrile in 13 min). Experiments were run in triplicates and protonated buffer was injected between each triplicate to wash the column and avoid cross over contamination. Peptide identification was performed with ProteinLynx Global SERVER (PLGS, Waters, Manchester, UK) based on the MSE data acquired on the non-deuterated samples. Peptides were filtered in DynamX 3.0 with the following parameters: peptides identified in at least 3 out of 5 acquisitions, 0.3 fragments per amino-acid, intensity threshold 1000. The Relative Deuterium Uptakes were not corrected for back exchange. Deuteros 2.0 software[52] was used for data visualization and statistical analysis. The online web application HDX-Viewer[53] and PyMOL (The PyMOL Molecular Graphics System, Version 2.3.0 Schrödinger, LLC) were used to represent the HDX-MS data on either the LptDE and LptDEM models.

The native MS and HDX-MS data has been deposited to the ProteomeXchange Consortium via the PRIDE partner repository[54] with the dataset identifier PXD041774. A description summary of the HDX data[55] is reported in Supplementary Table 2.

## LPS extraction and silver staining

*E. coli* strains were cultured till OD600 = 0.5. 1.5 ml aliquots of cultures were withdrawn to harvest cells by centrifugation. LPS was extracted following an hot aqueous-phenol extraction[56]. Briefly, cell pellets were resuspended in 200 μl blue buffer (50 mM Tris-HCl pH 6.8, 2% (v/v) β-mercaptoethanol, 2% (w/v) SDS, 10% glycerol and a pinch of bromophenol blue). After boiling for 15 min and cooling at room temperature for a similar time, samples were treated with 0.5 mg ml$^{-1}$ proteinase K overnight at 59 °C. A first extraction was conducted by applying to the sample 200 μl of ice-cold water-saturated phenol and heating at 65 °C for 15 min with agitation. After cooling to room temperature, 1 ml diethyl ether was applied followed by agitation by vortexing for 15 s and centrifugation at 20,600 x *g*. The heavier blue LPS-containing phase was withdrawn and ~1/10 of the obtained LPS fraction was resolved by SDS-PAGE before direct visualization by Silver staining (SilverQuest Silver Stain, Invitrogen).

## Gel electrophoresis Coomassie staining and Western blotting

Proteins samples were prepared in Laemmli sample buffer (Bio-Rad) lacking β-mercaptoethanol (non-reducing conditions) or supplemented with β-mercaptoethanol (reducing conditions). Proteins were separated by home-made SDS gels (10% acrylamide in Bis-Tris pH 6.4 buffer, subjected to electrophoresis using either MES or MOPS buffer) or home-made blue native 6-13% acrylamide gradient gel[49]. Where indicated, gels were stained with Coomassie brilliant blue R250. The signal intensities of protein bands were quantified using the Multi Gauge software (Fujifilm). To perform Western blots, protein gels were blotted onto PVDF membranes (Merck). Upon blocking with skim milk, membranes were incubated with epitope-specific rabbit polyclonal antisera or with an anti-polyhistidine horseradish peroxidase-conjugated monoclonal antibody (TaKaRa). Immunodetection was revealed by using a Clarity Western ECL blotting substrate (BioRad) and detected using a LAS 4000 (Fujifilm) apparatus. Antisera were raised in rabbits against peptides or full proteins from Escherichia coli (LptM, dilution 1:1000; BamA, 1:1000; BamD, 1:1000; BamE, 1:1000; SurA, 1:1000; Skp, 1:1000; OmpA, 1:1000; CyoA, 1:1000; LptD, 1:5000; LptE, 1:10000; LptA, 1:1000; LptB, 1:10000) or Saccharomyces cerevisiae (F1beta, 1:1000). Horseradish peroxidase conjugated anti-polyhistidine (TaKaRa product n. 631210, 1:2000) and horseradish peroxidase conjugated anti-rabbit IgG (Sigma product n. A6154, 1:10000) were purchased. The rabbit polyclonal antiserum against LptD was a gift of Dr. J.F. Collet (UC Louvain, Belgium) and the rabbit polyclonal antisera against LptE, LptA and LptB were gifts of Dr. A. Polissi (University of Milan, Italy).

## Protein model building

Models of LptDE and LptDEM were built using AlphaFold2, running AlphaFold Multimer v1.0 via ColabFold[35–37]. Models were built using the *E. coli* sequences: residues 26-784 of P31554 for LptD, residues 19-193 of P0ADC1 for LptE, and residues 20-67 of P0ADN6 for LptM (previously YifL). Five rounds were run for each prediction, with the highest ranked model used for follow-up analysis. All AlphaFold2 LptDE and LptDEM models are available for download at https://osf.io/xpfjc/, along with PAE, coverage and pIDDT plots. Note that the C-terminus of LptE was truncated after Thr174 for all images used in the manuscript.

## Site-directed photocrosslinking

Δ*lptM* cells were co-transformed with pEVOL-pBpF and plasmids expressing Lpt components with *p*Bpa at the indicated positions (see Figure Legends and Table 2). Cells transformed with pLptM$^{His}$ and LptDME$^{His}$ derivative plasmids harboring amber codons respectively in *lptM* and *lptE* ORFs were cultured in M9 minimal media, whereas cells transformed with pLptDEM$^{His}$ harboring amber codons in the *lptD* ORF were cultured in LB. At mid-exponential phase, cells were supplemented with 1 mM *p*Bpa and 200 μM IPTG for 1.5 h. Two identical cell culture aliquots were withdrawn, and one was kept on ice protected from light, whereas the other was subjected to UV irradiation (Tritan 365 MHB, Spectroline) for 10 min on ice. Envelope fractions were

prepared from both non-treated and UV irradiated cells. To monitor the crosslinks adducts of *p*Bpa in LptM or LptE, cells were solubilized with 20 mM Tris-HCl, pH 8, 12% (w/v) glycerol, 4% (w/v) SDS, 15 mM EDTA and 2 mM PMSF. After removing non-solubilized material by centrifugation, the supernatants were either directly resolved by SDS-PAGE (Load) or diluted in RIPA buffer (50 mM Tris/HCl pH 8, 150 mM NaCl, 1% [v/v] NP-40, 0.5% [w/v] sodium deoxycholate, 0.1% [w/v] SDS) and subjected to Ni-affinity purification as previously described[49] (Elution). To monitor the crosslink adducts of *p*Bpa in the population of LptD associated to LptM$^{His}$, we solubilized the envelope fraction with DDM and conducted Ni-affinity purification under native conditions as described above.

### Molecular dynamics simulations

The top ranking AlphaFold2 models for LptDE and LptDE-LptM were used to seed MD simulations. The models were built into simulation systems using CHARMM-GUI[57,58]. Protein atoms were described with the fixed charge CHARMM36m force field[59,60]. The C-terminus of LptE was truncated after Thr174 to remove contacts with the LptD barrel, and because the lDDT was low for this region. The N-terminal cysteine residues of LptM and LptE were tri-palmitoylated. Side chain pKas were assessed using propKa3.1[61], and side chain side charge states were set to their default, apart from Glu263 and Asp266 of LptD, and Lys103 of LptE, which were all set to neutral. The proteins were built into asymmetric membranes, comprising 6:3:1 palmitoyl-oleoyl phosphatidylethanolamine (POPE), palmitoyl-oleoyl phosphatidylglycerol (POPG), and cardiolipin in the inner leaflet, with LPS in the outer leaflet. The *E. coli* Lipid A core with two 3-deoxy-α-D-manno-octulosonic acid units was chosen as a representative LPS molecule. The membranes were solvated with TIP3P waters and neutralised with K⁺, Cl⁻ and Ca²⁺ to 150 mM. This choice of force field and water model should be sufficient to model the internal protein dynamics needed for the study[59]. System boxes were ~11 × 11 x 13.5 nm, with ~170,000 atoms (see Supplementary Table 3). Each system was minimized and equilibrated according the standard CHARMM-GUI protocol. Production simulations were run in the NPT ensemble, with temperatures held at 303.5 K using a velocity-rescale thermostat and a coupling constant of 1 ps, and pressure maintained at 1 bar using a semi-isotropic Parrinello-Rahman pressure coupling with a coupling constant of 5 ps[62,63]. Short range van der Waals and electrostatics were cut-off at 1.2 nm. Simulations were run to 500 ns and in triplicate for each system using independent initial starting velocities.

All simulations were run in Gromacs 2020.1 (https://doi.org/10.5281/zenodo.7323409)[64]. Data were analyzed using Gromacs tools (including residue distances, H-bond number, RMSF and angle analysis) and visualized in VMD[65]. Plots were made using Matplotlib https://ieeexplore.ieee.org/document/4160265.

### Reporting summary

Further information on research design is available in the Nature Portfolio Reporting Summary linked to this article.

## Data availability

Sources data are provided within this paper. The genomes used in this studies, their accession codes and corresponding hyperlinks are listed in the Source Data file. MALDI-TOF/TOF data and quantifications of coomassie-stained gel protein bands in Fig. 3d are available in the Source Data file. Un-cropped gels are available in Supplementary Figs. 14 and 15 of the Supplementary Information file. Native- and HDX-MS data are available via the PRIDE partner repository under the accession code PXD041774. Structural models are available via the Open Science Framework under the accession code xpfjc. Source data are provided with this paper.

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

## Acknowledgements

Research in R.I.'s lab was funded by the CNRS (ATIP-Avenir grant to R.I.), the China Scholarship Council fellowships to Y.Y. and H.C. and the FRM postdoctoral fellowship to D.R. HDX-MS experiments were supported by the French Ministry of Research (Investissements d'Avenir Program, Proteomics French Infrastructure, ANR-10-INBS-08 to O. B.-S.) and the Région Midi Pyrénées to O.B.-S. Research in P.J.S.'s lab was funded by Wellcome (208361/Z/17/Z) and BBSRC (BB/P01948X/1, BB/R002517/1 and BB/S003339/1). This project made use of time on ARCHER2 and JADE2 granted via the UK High-End Computing Consortium for

Biomolecular Simulation, HECBioSim (http://hecbiosim.ac.uk), supported by EPSRC (grant no. EP/R029407/1). This project also used Athena and Sulis at HPC Midlands+, which were funded by the EPSRC on grants EP/P020232/1 and EP/T022108/1. We thank the LMGM, Toulouse, for access to the *E. coli* Keio collection. We thank the IPBS, Toulouse, for access to the MALDI-TOF/TOF mass spectrometer. We thank the University of Warwick Scientific Computing Research Technology Platform for computational access.

## Author contributions

Y.Y. and R.I. conceived the research. Y.Y., H.C., V.M., D.R. and RI performed the genetic and biochemical analyses; R.A.C. and P.J.S. performed structural modeling and molecular dynamics simulations; Y.Q. performed phylogenetic analyses; J.M. performed native-MS and HDX-MS analyses with the help of C.F.; A.C.-S. performed the MALDI-TOF analysis with the help of C.A.; Y.Y., H.C., R.A.C., V.M., Y.Q., A.C.-S., J.M. and R.I. prepared the figures; R.A.C., Y.Q., P.J.S. J.M. and R.I. wrote the manuscript with contributions from all other authors; Y.Q., P.J.S., J.M. and R.I. supervised the work; O.B.-S., P.J.S., J.M. and R.I. acquired funding.

## Competing interests

The authors declare no competing interests.
