## [Peer Review File · Nature Communications]

Reviewers' Comments:

Reviewer #1:

Remarks to the Author:

The manuscript by Yang and co-workers identifies and characterizes LptM, (formerly YifL), as an outer membrane (OM) lipoprotein that binds the OM LPS translocon LptDE and promotes its oxidative maturation. The authors show that LptM co-purify with LptDE and when the three proteins (LptDEM) are overexpressed they form a 1:1:1 heterotrimeric complex. While lptM deleted mutants do not display significant OM permeability defects, the absence of LptM seems to slow down LptDE assembly at the OM by the BAM complex as judged by co-purification of BAM components only in lptM deleted cells.

The mature form of LptD possesses two disulfide bonds formed by non-consecutive pairs of Cys residues whose correct formation is crucial for the assembly of a functional LptDE translocon. The authors implicate LptM in oxidative LptD maturation by showing accumulation of a partially oxidized form of LptD containing non-native disulfide bonds in lptM deleted cells and the requirement of DsbC to isomerize the non-correctly formed disulfide bonds. Finally, based on molecular dynamics simulation studies, the authors suggest that the association of LptM to LptD stabilizes the OM translocon in a way similar to that observed by LPS binding. Based on this final observation the authors extend the role of LptM as the eighth component of the Lpt protein machinery not only implicated in LptDE oxidative maturation but also actively engaged in LPS transport.

The work is solid, well written and well presented. The identification of LptM as a crucial factor in LptDE maturation and assembly adds a very important piece of information to the LPS biogenesis pathway. Compelling evidence implicates LptM in LptD oxidative maturation. However, I am not fully convinced that LptM can be considered as an additional structural component of the Lpt machinery (please see my few specific comments below).

Specific comments

Figure 2 panel B

The level of LptD purified from wild type and lptM deleted cells harboring pLptDE-His should be shown. Why BamA protein is not co-purified along with BamD and BamE?

Figure 2 panel C

LptM overexpressing cells co-purify BamA, BamD and BamE along with LptD. Another interpretation of these results can be that LptM also binds components of the Bam complex. Is overexpression of LptM toxic thus impacting on LptDE residence on Bam complex? In wild type cells ectopically expressing LptDE Bam components are not copurified (panel B line 5).

Could you please comment?

In my opinion these data support a model in which LptM is essentially implicated in LptDE oxidative maturation but not in LPS transport. To prove that LptM is a true component of the Lpt machinery the authors should co-purify it with the entire Lpt trans-envelope complex possibly using a different bait (e.g. LptC or LptB) and in conditions in which LptDE are not overexpressed.

Discussion

The authors should comment/discuss BepA, a protease/chaperone that facilitates disulfide bonds isomerization in LptD. BepA binds LptD and BamA; in the absence of BepA cells accumulate LptD with non-native disulfide bonds.

The authors assign LptM an important regulatory function in activating the LptDE translocon by mimicking substrate binding. This hypothesis is mainly drawn by MD simulation and HDX-MS studies. However, unlike all other seven Lpt components, lptM is not essential and its deletion does not affect OM permeability suggesting that LPS export is minimally affected. Even the regulatory component LptC, which controls the ATPase activity of LptB2FG ABC transporter is essential in the absence of suppressor mutations.

Can LptM be rather viewed as an important factor required to guarantee correct LptDE maturation and therefore to ensure optimal LPS transport under all growth conditions?

Minor comments

Page 8: "Figure 2A panels 3-6" should be "Figure 2A lanes 3-6"

Figure 3 panel C: please indicate to which LptD species filled and empty arrows refer to

Figure 4 panel D: I suppose that in the right panel the different LptD variants are overexpressed (by IPTG), this should be indicated in the figure

Reviewer #2:

Remarks to the Author:

The authors describe the identification of a new protein component of the LPS translocon, LptM. In a series of measurements, they find that LptM stabilizes the LptD complex so that that it can form its 2 disulfide bonds and then be released as the native LPS translocon by the BAM complex. The conclusion that LptM interacts with LptD/E is convincing and of interest. My main concern is that the position of LptM in the proposed LptD/E/M model is not strongly supported experimentally or by the AF2 analysis itself. It may turn out to be correct, but the authors need to do a more thorough analysis (photo-crosslinking and point mutations) to support the model. I support publication subject to addressing the modeling and HDX-mass spectrometry issues described below.

Modeling:

1. The photocrosslinking data shown in Fig S4 are complex. It is important to further validate the assignments using antibodies against LptE and LptD, as is done for OmpA
2. Did the authors look for LptM crosslinks to the LptE subunit? Based on the AF2 modeling, one would expect to see additional crosslinks to LptE and LptD and the LptM C-terminal region.
3. The AF2 PAE plots seem to indicate that placement (and conformation) of LptM is rather low confidence. This is especially true for the C-terminal region, even though in the proposed model it is observed to make multiple contacts with LptD and LptE subunits. Does this explain why sites in the LptM C-terminal region are not observed to crosslink to LptE or LptD?
4. The authors provide a useful link to download the relevant AF2 files, but they should also include some of this information as a supplementary figure to make it easily accessible to the reader (e.g., PAE matrices; models colored by pLDDT).
5. The authors should include a figure showing sequence conservation mapped to their proposed LptD/E/M complex—are the predicted interaction surfaces with LptM enriched in conserved residues?
6. Fig 5—clarify by using different colors to indicate HX protection vs. LptM subunit. Also, why is LptM shown as spheres rather than Ca trace (cartoon) as for the other subunits?
7. On p 14 the authors write that the "top-ranking structures reveal a considerable amount of contact between LptM and LptDE (Figure 5A), with the N-terminal cysteine of LptM consistently placed in close proximity of the flexible hinge separating the periplasmic and membrane domains of LptD that is adjacent to the LptD b-barrel lateral gate (Figure S5A), which is consistent with our crosslinking data (Figure S4A and B)." This should be tightened up—while the crosslinking analysis indeed shows that the N-terminal region is close to LptD, it says nothing about where it interacts with the large LptD subunit.

HD exchange-Mass Spectrometry.

1. Is there a reason why the authors did not correct for back exchange? This could affect the interpretation of the effects of adding LptM. Minimally, the value of the all-D control should be noted on the deuterium build-up curves so that extent of deuteration can be assessed.
2. That none of the build-up curves shown go above ~50% D labeling is not ideal (Fig. S10). Furthermore, the time-dependent deuterium uptake curves are very flat, essentially only showing vertical shifts upon the addition of LptM. Are uptake kinetics actually observed, especially for any peptides that end up highly deuterated? Generally, this reviewer cannot tell if the data are supporting a model where the addition of LptM slows exchange across the whole peptide (i.e., large scale stabilization), or LptM addition just results in the additional protection of a few residues which occurs in the deadtime of the measurement, reflected as a vertical shift before 0.4 minutes,

but otherwise has no effect on the rest of the residues. A vertical shift would still be very useful information, and possibly sufficient for their conclusions, but it needs to be properly interpreted.

3. It would be good if the authors showed the time course for the critical peptides for their conclusions rather than just those that exhibit bimodality.

4. The authors state that they observe EX1 behavior. This isn't necessarily true. Bimodal mass spectra can occur under true EX1 conditions when closing rates are slower than the intrinsic chemical exchange rates. But they can also occur when there are two populations that don't interconvert during the labeling time period. A further condition for true EX1 behavior is that the sum of two areas of the two peaks remain constant (one peak goes down while the other goes up). An examination of the mass spectra in Fig. S10 suggests that there are two non-interconverting populations rather than true EX1 behavior. This should be noted in the revised ms.

5. This reviewer appreciates the authors' deposition of the HDX data. However, they could also present a table describing the data as recommended in "Recommendations for performing, interpreting and reporting hydrogen deuterium exchange mass spectrometry (HDX-MS) experiments". <https://www.nature.com/articles/s41592-019-0459-y#Tab1>

REVIEWER COMMENTS

Reviewer #1 (Remarks to the Author):

The manuscript by Yang and co-workers identifies and characterizes LptM, (formerly YifL), as an outer membrane (OM) lipoprotein that binds the OM LPS translocon LptDE and promotes its oxidative maturation. The authors show that LptM co-purify with LptDE and when the three proteins (LptDEM) are overexpressed they form a 1:1:1 heterotrimeric complex. While lptM deleted mutants do not display significant OM permeability defects, the absence of LptM seems to slow down LptDE assembly at the OM by the BAM complex as judged by co-purification of BAM components only in lptM deleted cells. The mature form of LptD possesses two disulfide bonds formed by non-consecutive pairs of Cys residues whose correct formation is crucial for the assembly of a functional LptDE translocon. The authors implicate LptM in oxidative LptD maturation by showing accumulation of a partially oxidized form of LptD containing non-native disulfide bonds in lptM deleted cells and the requirement of DsbC to isomerize the non-correctly formed disulfide bonds. Finally, based on molecular dynamics simulation studies, the authors suggest that the association of LptM to LptD stabilizes the OM translocon in a way similar to that observed by LPS binding. Based on this final observation the authors extend the role of LptM as the eighth component of the Lpt protein machinery not only implicated in LptDE oxidative maturation but also actively engaged in LPS transport.

The work is solid, well written and well presented. The identification of LptM as a crucial factor in LptDE maturation and assembly adds a very important piece of information to the LPS biogenesis pathway. Compelling evidence implicates LptM in LptD oxidative maturation. However, I am not fully convinced that LptM can be considered as an additional structural component of the Lpt machinery (please see my few specific comments below).

1) We thank the reviewer for her/his appreciation of our study. As highlighted by the reviewer, the main message of our study is that LptM promotes oxidative maturation of the LPS translocon. Our structural prediction, molecular dynamics simulations and HDX-MS analysis also suggest that LptM interacts with different domains of the LPS translocon that have been proposed to coordinate LPS insertion into the OM. The new experiments presented in the revised manuscript further validate the structural prediction and provide new evidence that LptM is a true component of the Lpt transenvelope pathway (see answers to the specific comments below).

Specific comments

Figure 2 panel B

The level of LptD purified from wild type and lptM deleted cells harboring pLptDE-His should be shown.

2) *As requested by the reviewer, we have enlarged the gel crop in the modified Fig. 2b of the revised manuscript (lanes 1 and 2, before showing only LptE^{His}) to show also the levels of LptD.*

Why BamA protein is not co-purified along with BamD and BamE?

3) *BamA and BamDE can exist in distinct functional submodules of the BAM complex, that are BamAB and BamCDE, respectively. These submodules can assemble together to form a functional BAM complex (Hagan et al., 2010, Science 328:890-2). It is thus conceivable that the LptDE folding intermediate accumulating in the absence of LptM interacts most stably with the BamDE-containing submodule of the BAM complex.*

Figure 2 panel C

LptM overexpressing cells co-purify BamA, BamD and BamE along with LptD. Another interpretation of these results can be that LptM also binds components of the Bam complex. Is overexpression of LptM toxic thus impacting on LptDE residence on Bam complex? In wild type cells ectopically expressing LptDE Bam components are not copurified (panel B line 5). Could you please comment?

4) *As highlighted by the reviewer, LptM binds to some degree to the BAM complex. We show this result in Supplementary Fig. 2a of the revised manuscript (corresponding to Fig. 2C of the previous version of our manuscript). We believe this interaction reflects the fact that LptM is assembled together with LptDE at the BAM complex. Currently, the exact order of events that brings LptM, LptD and LptE to the BAM complex (e.g. whether LptM binds the BAM complex prior to LptDE) are not known. Therefore, it is not possible to provide any detailed explanation as to why BAM subunits are co-purified to some extent with LptM, whereas BAM is less efficiently copurified with LptDE.*

In any cases, we show in Fig. 3d (lanes 1 and 2) that the overproduction of LptM concomitant to the overproduction of LptDE has a beneficial effect on LptD assembly, enhancing the efficiency of LptD oxidative maturation. Thus, rather than hampering LptD assembly at the BAM complex, LptM promotes LptD oxidative maturation. Furthermore, following the suggestion of the reviewer, we have tested the effect of LptM overproduction by a drop dilution test. Our results (new Supplementary Fig. 2b, left, of the revised manuscript) show that the overproduction of LptM does not cause any detectable toxicity. In addition to the reviewer request of testing whether LptM overproduction is not toxic, we also verified that LptM overproduction does not interfere with the outer membrane permeability barrier to the large molecular weight antibiotic vancomycin (Supplementary Fig. 2b, right).

In my opinion these data support a model in which LptM is essentially implicated in LptDE oxidative maturation but not in LPS transport. To prove that LptM is a true component of the Lpt machinery the authors should co-purify it with the

entire Lpt trans-envelope complex possibly using a different bait (e.g. LptC or LptB) and in conditions in which LptDE are not overexpressed.

5) *We expanded our previous analysis of LptM^{His} purification to test whether LptM^{His} co-purifies other Lpt components in addition to LptDE, which are not overexpressed. Our new western-blots (added to the new Supplementary Fig. 2a of the revised manuscript) reveal that LptA and LptB are also present in the elution of LptM^{His}. Together with the results that LptM-associated LptD can be crosslinked to LptA (see our response point 7 to this reviewer) our analysis provides some evidence that LptM integrates the Lpt transenvelope pathway. We highlight this new important finding in the Results section (page 7) “We conclude that LptM stably interacts with the OM LPS translocon LptDE, thus integrating the transenvelope Lpt pathway”*

Discussion

The authors should comment/discuss BepA, a protease/chaperone that facilitates disulfide bonds isomerization in LptD. BepA binds LptD and BamA; in the absence of BepA cells accumulate LptD with non-native disulfide bonds.

6) *We agree with the reviewer that it is relevant to consider the role of BepA as this chaperone/protease plays a role during an early step of LptD folding at the BAM complex. In addition to highlighting the function of BepA as requested by the reviewer, we have expanded our analysis to verify whether LptM plays a role in LptD assembly that is synergistic with that of BepA. To this end, we have generated new strains lacking only BepA or both LptM and BepA and analyzed LptD oxidation intermediates by Western blotting. Consistent with previous studies, (Weski and Ehrmann, J. Bacteriol 2012, 194:3225-33; Soltes et al., J. Bacteriol 2017, 199:e00418-17), the deletion of bepA does not influence the steady-state level of LptD (new Fig. 3c lanes 13-16 of the revised manuscript). Similarly, the deletion of bepA in Δ lptM does not alter the diminished level of oxidized LptD and the accumulation of the oxidation intermediate LptD^{C1-C2} that was already obtained with the deletion of only lptM. The data suggest that BepA does not function synergistically with LptM nor degrades LptD assembly intermediates in delta lptM, under our experimental conditions.*

The authors assign LptM an important regulatory function in activating the LptDE translocon by mimicking substrate binding. This hypothesis is mainly drawn by MD simulation and HDX-MS studies. However, unlike all other seven Lpt components, lptM is not essential and its deletion does not affect OM permeability suggesting that LPS export is minimally affected. Even the regulatory component LptC, which controls the ATPase activity of LptB2FG ABC transporter is essential in the absence of suppressor mutations. Can LptM be rather viewed as an important factor required to guarantee correct LptDE maturation and therefore to ensure optimal LPS transport under all growth conditions?

7) *The reviewer raises a key point, that is whether LptM plays a role only in LptDE biogenesis or also as a regulator of lipopolysaccharide transport. As stated by the reviewer, we clearly show that LptM has a role in the oxidative maturation of LptDE and that LptM is essential when DsbC is inactivated. Based on these results, we decided to rename this protein (formerly YifL) as LptD oxidative Maturation associated lipoprotein or LptM.*

In addition, our biochemical and structural analyses also show that LptM can stably associate with the mature LptDE complex, as well as with periplasmic and inner membrane components of the Lpt pathway (see response point 5 to this reviewer). Inspired by the comments of this reviewer, we have addressed whether LptM remains at the LptDE translocon when the latter interacts with other components of the transenvelope Lpt pathway. To this end, we assessed whether the population of LptD that stably interacts with LptM can associate with LptA. We introduced the photoactivatable amino acid pBpa in place of LptD Y63, a position that was shown to interact with LptA in vivo (Freinkman et al., PNAS 2011, 108:2486-91). Our results (new Fig. 1d of the revised manuscript) reveal that a significant fraction of LptD co-isolated along with LptM had been crosslinked to LptA in vivo, prior to the purification step. These results indicate that LptM is part of the functional LptDE translocon that interacts with LptA. Together with our structural prediction, MD simulation and HDX-MS studies, our new findings further support the hypothesis that LptM may play an important regulatory function at the Lpt transenvelope pathway. Nevertheless, in the Discussion (page 21) we clarify that further studies will be warranted to test this hypothesis: "A possible scenario, which will warrant further analyses, is that LPS substitutes LptM in the membrane-embedded portion of the translocon prior to breaching the OM".

Minor comments

Page 8: "Figure 2A panels 3-6" should be "Figure 2A lanes 3-6"

8) *We have improved the reference to images of the drop dilution tests in Fig. 2a by replacing the term "panel" with "frame". We prefer not to use the word "lane" to avoid any confusion with the lanes of gels.*

Figure 3 panel C: please indicate to which LptD species filled and empty arrows refer to

9) *We have added this description to our Figure Legend.*

Figure 4 panel D: I suppose that in the right panel the different LptD variants are overexpressed (by IPTG), this should be indicated in the figure

10) *We thank the reviewer for identifying this omission in our figure labeling, which has now been corrected in Fig. 4d.*

Reviewer #2 (Remarks to the Author):

The authors describe the identification of a new protein component of the LPS translocon, LptM. In a series of measurements, they find that LptM stabilizes the LptD complex so that that it can form its 2 disulfide bonds and then be released as the native LPS translocon by the BAM complex. The conclusion that LptM interacts with LptD/E is convincing and of interest. My main concern is that the position of LptM in the proposed LptD/E/M model is not strongly supported experimentally or by the AF2 analysis itself. It may turn out to be correct, but the authors need to do a more thorough analysis (photo-crosslinking and point mutations) to support the model. I support publication subject to addressing the modeling and HDX-mass spectrometry issues described below.

Modeling:

1. The photocrosslinking data shown in Fig S4 are complex. It is important to further validate the assignments using antibodies against LptE and LptD, as is done for OmpA

1) We have improved the presentation of our crosslink data (Fig. S4 of the original manuscript submission). First the LptM photocrosslink analysis is now presented in Supplementary Fig. 8 together with LptE crosslink analysis as both sets of results provide a solid support to our structural prediction. We had previously shown that the UV-induced 100 kDa crosslink product obtained for position LptM L22 contains disulfide bonds, from which we inferred the presence of LptD (Fig. S4B of the original submission). In a new experiment, by using both LptM and LptD antibodies we fully validate the identification of the UV-induced adduct of 100 kDa as LptM-LptD crosslink. This new analysis is shown in the new Supplementary Fig. 8c of the revised manuscript, which replaces the former results of Fig. S4B.

2. Did the authors look for LptM crosslinks to the LptE subunit? Based on the AF2 modeling, one would expect to see additional crosslinks to LptE and LptD and the LptM C-terminal region.

2) We were unable to identify crosslink products of LptM C-terminal region to LptE or LptD. It should be noted that this region of lptM is highly unstructured and therefore it is likely to mediate multiple short-lived interactions, which would hamper crosslink efficiency. In addition, our MD simulations show that the LptM C-terminal region is highly dynamic, probably coordinating multiple interactions. We have modified our text in the Results (pages 14-15) to highlight this feature of LptM: "It should be noted, however, that our MD simulations predicted that the highly unstructured LptM C-terminal region is conformationally dynamic (Supplementary Fig. 6c), suggesting that it may coordinate multiple interactions".

To obviate the limitation of our LptM photocrosslink approach, we have expanded our analysis by probing LptE with pBpa. Our main objective was to gain further evidence that LptM occupies the lumen of the LptD beta-barrel in proximity of

LptE. To this end, pBpa was introduced at different amino acid positions in LptE: K70 faces the periplasm oriented toward LptM in our structural model; V80, I82, and A83 align along a beta-strand of LptE that faces the lumen of the LptD barrel occupied by the LptM C-terminal region; K121 is positioned in a beta-strand of LptE distal from the lumen of the barrel left empty by LptE. We found that K70, V80, I82 and A83 were crosslinked to LptM with different efficiencies, whereas LptE K121 did not form any crosslink product. These results are consistent with our prediction that LptM occupies a portion of the lumen of LptD left empty by LptE. This analysis (new Supplementary Fig. 8e of the revised manuscript) further supports our structural prediction.

3. The AF2 PAE plots seem to indicate that placement (and conformation) of LptM is rather low confidence. This is especially true for the C-terminal region, even though in the proposed model it is observed to make multiple contacts with LptD and LptE subunits. Does this explain why sites in the LptM C-terminal region are not observed to crosslink to LptE or LptD?

3) We agree with the reviewer that the placement of LptM C-terminal region is low confidence. Multiple factors are likely to weaken this score. First, this region of LptM is unstructured and poorly conserved in its amino acid sequence. Second, our MD simulations show that indeed this region of LptM is more conformationally dynamic than the LptM N-terminus. The short-lived contact of LptM within the lumen of the barrel probably explain inefficient crosslink when pBpa is introduced in LptM. As stated above, we have amended our text at pages 14-15 by highlighting these features of LptM C-terminal region: "It should be noted, however, that our MD simulations predicted that the highly unstructured LptM C-terminal region is conformationally dynamic (Supplementary Fig. 6c), suggesting that it may coordinate multiple interactions".

4. The authors provide a useful link to download the relevant AF2 files, but they should also include some of this information as a supplementary figure to make it easily accessible to the reader (e.g., PAE matrices; models colored by pLDDT).

4) The PAE matrices and pLDDT along with LptDEM sequence coverage are now shown in the new Supplementary Fig. 5a-c of the revised version of our manuscript.

5. The authors should include a figure showing sequence conservation mapped to their proposed LptD/E/M complex—are the predicted interaction surfaces with LptM enriched in conserved residues?

5) Following the suggestion of the reviewer, we now highlight sequence conservation of predicted interaction sites on the structure of LptDEM. This representation, shown in the new Supplementary Fig. 7 of the revised manuscript, highlights that the interaction sites of LptM N-terminal region with LptD are

conserved in Enterobacteriaceae, although sequence conservation decreases towards the C-terminal region of LptM. The procedure to prepare this new figure has been added to the Materials and Methods section.

6. Fig 5—clarify by using different colors to indicate HX protection vs. LptM subunit. Also, why is LptM shown as spheres rather than Ca trace (cartoon) as for the other subunits?

6) We have changed the color of LptM (to purple) in Fig. 5e and all other panels illustrating the structure of LptM. We represent LptM as spheres to better distinguish this small protein from the larger LptD and LptE partners represented as cartoon.

7. On p 14 the authors write that the “top-ranking structures reveal a considerable amount of contact between LptM and LptDE (Figure 5A), with the N-terminal cysteine of LptM consistently placed in close proximity of the flexible hinge separating the periplasmic and membrane domains of LptD that is adjacent to the LptD b-barrel lateral gate (Figure S5A), which is consistent with our crosslinking data (Figure S4A and B).” This should be tightened up—while the crosslinking analysis indeed shows that the N-terminal region is close to LptD, it says nothing about where it interacts with the large LptD subunit.

7) We agree with the reviewer and we made the suggested change by removing the part of the sentence “which is consistent with our crosslinking data (Fig. S4A and B)”. We now highlight, more precisely in a separate sentence, that the interaction of LptM N-terminal region with LptD is supported by our crosslinking data. As described above we provide further experimental support to the predicted model by showing that LptE in the lumen of the LptD beta-barrel can crosslink LptM. We have collected these crosslink analyses together in Supplementary Fig. 8b-e of the revised manuscript.

HD exchange-Mass Spectrometry.

1. Is there a reason why the authors did not correct for back exchange? This could affect the interpretation of the effects of adding LptM. Minimally, the value of the all-D control should be noted on the deuterium build-up curves so that extent of deuteration can be assessed.

8) In our study, we mainly used HDX-MS to compare the deuterium incorporation of LptDE vs. LptDEM. Even though the reviewer is right to say that the level of back-exchange can be different for each peptide, the back-exchange of each peptide should be the same in both conditions, so the differential comparison (Figs. 5d,e and Supplementary Figs. 10 and 11 of the revised manuscript) should not be affected. However, we also represent in Supplementary Figure 9 of the revised manuscript the relative fractional uptake of LptDE, so we conducted the proposed experiment by fully deuterating LptDEM in 8M d4-Urea for 24h at 20°C prior to LC-MS analysis. This procedure was added to the Materials and Methods section and a new PXD repository was generated to include this maximally

deuterated control. Reviewers can access it using the account details: Username: reviewer_pxd041774@ebi.ac.uk ; Password: Yk50qwcB

A slight shift in the retention time led to the loss of 15 peptides (out of 155) but we could preserve identical sequence coverages for both LptD and LptE. Supplementary Figs. 9-11 of the revised manuscript were modified accordingly and, as recommended by the reviewer, these values were added on the deuteration kinetics (new Supplementary Fig. 12 and Supplementary Files 1 and 2 of the revised manuscript). In some cases, the value of this maximally deuterated control is identical to the value obtained after 30min deuteration: this is specified in the figure legends.

Back-exchange values were calculated for each peptide: they range from 35,1% to 79.9%, with an average value of 54.9%; but only ~30% of the peptides have more than 60% back-exchange. The back-exchange is more important to what is usually reported (20-40%) but arise from the fact that we chose to use a gradient longer than usual (15 min vs. 7 min), in order to maximize sequence coverages. These results are now presented in the new Supplementary Fig. 13 of the revised manuscript.

2. That none of the build-up curves shown go above ~50% D labeling is not ideal (Fig. S10). Furthermore, the time-dependent deuterium uptake curves are very flat, essentially only showing vertical shifts upon the addition of LptM. Are uptake kinetics actually observed, especially for any peptides that end up highly deuterated? Generally, this reviewer cannot tell if the data are supporting a model where the addition of LptM slows exchange across the whole peptide (i.e., large scale stabilization), or LptM addition just results in the additional protection of a few residues which occurs in the deadtime of the measurement, reflected as a vertical shift before 0.4 minutes, but otherwise has no effect on the rest of the residues. A vertical shift would still be very useful information, and possibly sufficient for their conclusions, but it needs to be properly interpreted.

9) *The maximum relative uptakes observed in this dataset were 53.0% for [25-32] of LptD and 56.6% for [156-164] of LptE. As mentioned previously, we agree that these relative deuterium uptakes are relatively low, but they are consistent with previous results obtained in our lab on other protein complexes (Guillet et al, 2019 Nat. Commun. 10:782 (max:56.84%), Lesne et al, 2020 Nat. Commun. 11:6140 (max:65.6%)), and most probably result from the back-exchange that is known to be quite important on the SynaptG2Si instrument, especially with our 15 min chromatographic gradient. Our additional experiments indeed estimated an average back-exchange of 54.9%. The maximally deuterated sample shows that the low deuteration observed on some peptides (e.g. [328-339], [576-584] of LptD, or [98-102], [148-155] of LptE) is clearly due to low exchange rates and not only to an extensive back-exchange.*

Concerning the "flatness" of the curves, the reviewer is right to say that the few uptake curves that were presented in Fig. S10 of the original submission (corresponding to Supplementary Fig. 12 of the revised version) were flat for the LptDE condition, and less so for the LptDEM condition. We are now including all

the uptake curves for LptD and LptE in the new Supplementary Files 1 and 2, respectively. Some of these uptake curves are not flat, in both LptD ([290-299], [533-540], [534-540], [664-671], [772-784], [772-784]) and LptE ([34-57], [34-59], [34-62], [36-57], [39-57], [98-102], [103-122], [107-119], [107-120], [107-122], [143-154], [143-155], [147-155], [148-155]).

In the peptides for which we see some significant differences, we actually see various cases:

- uptake curves parallel and flat (for eg. [198-203] in LptD or [78-89] in LptE)*
- uptake curves parallel and increasing ([212-235] in LptD or [34-59] in LptE)*
- flat curve for LptDE but increasing uptake for LptDEM (for eg. [212-218] in LptD or [127-134] in LptE). In some cases, the uptake of LptDEM reaches the one of LptDE (for eg. [70-82] in LptD).*
- increasing uptake for LptDE but a flat curve for LptDEM (for eg. [244-255] in LptD or [143-155] in LptE).*

From these results, it appears that the addition of LptM does not only result in the “early” protection of LptDE in the deadtime of the measurement (< 30 sec).

3. It would be good if the authors showed the time course for the critical peptides for their conclusions rather than just those that exhibit bimodality.

10) All the uptake plots are now shown in Supplementary Files 1 and 2.

4. The authors state that they observe EX1 behavior. This isn't necessarily true. Bimodal mass spectra can occur under true EX1 conditions when closing rates are slower than the intrinsic chemical exchange rates. But they can also occur when there are two populations that don't interconvert during the labeling time period. A further condition for true EX1 behavior is that the sum of two areas of the two peaks remain constant (one peak goes down while the other goes up). An examination of the mass spectra in Fig. S10 suggests that there are two non-interconverting populations rather than true EX1 behavior. This should be noted in the revised ms.

11) We thank the reviewer for the careful examination of our bimodal uptake plots. We agree that they do not exactly resemble those expected for true EX1 kinetics. We thus added the following sentence at the end of the results section: “However, a thorough analysis of the spectra does not show the typical EX1 behavior, whereby the first envelope decreases as the second increases, which could rather suggest the presence of two populations that do not interconvert during the labeling time period.” The term EX1 was also moved from the sentence “A bimodal isotopic distribution (EX1 regime) for peptides of the beta-taco domain was previously reported for Kp LptDE (Fiorentino et al., 2021)” to the sentence “Based on this result, it was suggested that the beta-strands of the beta-taco domain undergo concerted closing and opening motions that occur with a kinetic slower

than the rate of hydrogen/deuterium exchange (EX1 regime)” to better underline that this was the suggestion of the authors in the ref Fiorentino et al, 2021.

5. This reviewer appreciates the authors’ deposition of the HDX data. However, they could also present a table describing the data as recommended in “Recommendations for performing, interpreting and reporting hydrogen deuterium exchange mass spectrometry (HDX-MS) experiments”. <https://www.nature.com/articles/s41592-019-0459-y#Tab1>

12) This is a very good suggestion and the following Table is now presented in Table 4:

Table 4 : HDX data summary		
Dataset	LptDE	LptDEM
HDX reaction details	95% D ₂ O, pH 2.3, 10°C	
HDX time course (min)	0, 0.5, 1, 5, 10, 30	
HDX controls	Maximally labelled control in 8M d ₄ -urea in D ₂ O for 24h at 20°C / Blanks injected between each timepoint	
Back-exchange	54.4 ± 10.0 %	
Number of peptides	121 for LptD and 32 for LptE	
Sequence coverage	92.50% for LptD and 75.41% for LptE	
Average peptide Length / Redundancy	13.52/2.33 for LptD and 16.19/3.75 for LptE	
Replicates (technical)	3	
Repeatability (average SD)	0.0437 for LptD and 0.0850 for LptE	0.0492 for LptD and 0.1137 for LptE
Significant differences in HDX	hybrid significance test, p-value<0,001, Confidence Interval 0.52 Da for LptD and 1.04 for LptE	

Reviewers' Comments:

Reviewer #1:

Remarks to the Author:

In their revised manuscript Yang and co-authors addressed convincingly my concerns adding new experimental data.

The authors showed that LptM interacts with other members of the Lpt machinery, nevertheless I would be cautious in defining LptM as the eighth component of the Lpt complex (lines 511-512, of revised discussion) as this would imply that LptM is a core component with a role in LPS transport. LptM, unlike all other Lpt components, is not essential and its loss does not alter OM permeability suggesting that LPS transport is minimally affected. I would suggest rephrasing this sentence. I have no further questions.

Reviewer #2:

Remarks to the Author:

While the reviewers have responded well to most concerns including those related to hydrogen exchange, and the overall finding of LptM is significant, the author's model for the ternary complex—specifically, that the LptM C-terminal region binds within the LptD beta-barrel is unconvincing. They should provide more decisive experimental support—specifically, more rigorous site-specific cross-linking or mutational analysis, not just AF2 modeling for a disordered region nor molecular dynamics simulations. Otherwise, they should drastically tone down their argument that the LptM C-terminal domain contacts the lumen of the LptD beta-barrel.

Some details of the AF2 modeling are now shown in a new Sup Fig 5, but the requested panel showing the AF2 model colored by confidence is not provided. This is important because the top LptM model (indeed all five of the AF2 models) is actually very low confidence over nearly the entire protein except for the conserved N-terminus. This includes the C-terminal region that is modeled as if it is making extensive and important contacts within the center of the LptD barrel.

The authors attempt to provide additional support for the AF2 model of LptM by site-specific photocrosslinking. While these data are reasonable for the N-terminal interaction with LptD, unfortunately, they are unable to observe any pXL between the C-term of LptM and LptE or LptD when the benzophenone is incorporated into LptM. This concern is potentially mitigated because they claim to observe crosslinks to LptM when the benzophenones are incorporated into LptE (Sup Fig 8e). But these data are not particularly convincing. First, the authors once again fail to validate the higher MW species by blotting for LptE. Could the complex banding around 27-32 kDa be something else? Moreover, crosslinks are claimed to be seen with LptE K70 but not K121 (which serves here as a negative control), despite the latter being closer to the AF2-modeled position of LptM (and pointing towards it), and its location deep within the LptD barrel where it would presumably be less dynamic and more likely to crosslink.

The presentation of sequence conservation mapped to the structures of LptDEM (new Sup Fig 7) is confusing. Sequence conservation should be mapped to the structure of each component and then displayed in a manner that more clearly allows the reader to evaluate the proposed interaction surfaces.

REVIEWER COMMENTS

Reviewer #1 (Remarks to the Author):

In their revised manuscript Yang and co-authors addressed convincingly my concerns adding new experimental data.

The authors showed that LptM interacts with other members of the Lpt machinery, nevertheless I would be cautious in defining LptM as the eighth component of the Lpt complex (lines 511-512, of revised discussion) as this would imply that LptM is a core component with a role in LPS transport. LptM, unlike all other Lpt components, is not essential and its loss does not alter OM permeability suggesting that LPS transport is minimally affected. I would suggest rephrasing this sentence.

I have no further questions.

We thank the reviewer for the positive evaluation of our revised manuscript. Taking into account the warning of this reviewer, we have modified the beginning of our Discussion removing any emphasis on the number of Lpt subunits. Our newly revised Discussion starts with the sentence (lines 497-498) "Here we report the discovery of the lipoprotein LptM as a key factor for the oxidative maturation of the OM LPS translocon".

Reviewer #2 (Remarks to the Author):

While the reviewers have responded well to most concerns including those related to hydrogen exchange, and the overall finding of LptM is significant, the author's model for the ternary complex—specifically, that the LptM C-terminal region binds within the LptD beta-barrel is unconvincing. They should provide more decisive experimental support—specifically, more rigorous site-specific cross-linking or mutational analysis, not just AF2 modeling for a disordered region nor molecular dynamics simulations. Otherwise, they should drastically tone down their argument that the LptM C-terminal domain contacts the lumen of the LptD beta-barrel.

Some details of the AF2 modeling are now shown in a new Sup Fig 5, but the requested panel showing the AF2 model colored by confidence is not provided. This is important because the top LptM model (indeed all five of the AF2 models) is actually very low confidence over nearly the entire protein except for the conserved N-terminus. This includes the C-terminal region that is modeled as if it is making extensive and important contacts within the center of the LptD barrel.

We thank the reviewer for highlighting the relevance of our findings on the role of LptM in the biogenesis of the LPS translocon and acknowledging our revision work. The reviewer criticizes our emphasis on the possibility that the C-terminal segment of LptM can interact with translocon domains in the lumen

of the LptD barrel. It should be noted, however, that our results (including the new crosslinking experiment illustrated in the following response) clearly demonstrate that LptM forms a ternary complex with LptE and the LptD beta-barrel domain, and that segments of LptM occupy the internal lumen of the LptD beta-barrel.

As the C-terminal region of LptM is unstructured and highly dynamic, we agree with the reviewer to tone down the statements about the precise positioning of the LptM C-terminus, and in particular its interaction with the internal lumen of the barrel such as LptD loop4.

To do so, we introduced the following modifications to the Results and Discussion of the newly revised manuscript.

-Figure 5a: We have removed the insert highlighting the AlphaFold2 predicted interaction between the LptM C-terminal moiety and LptD loop 4 in the lumen of the beta-barrel domain.

-Results, lines 398-406: Whereas we still mention that LptM C-terminal region is predicted to interact with both LptE and LptD loop 4, we have removed the text describing specific possible salt bridges between LptM K53 and D55 and LptD loop 4 residues D330 and K346. We also mention the low prediction confidence score obtained specifically for the unstructured C-terminal region of LptM.

-New Supplementary Figure 5d: We plot the IDDT score on the predicted structure of the ternary complex LptDEM, as requested by the reviewer.

-Discussion, lines 517-519: We highlight the low confidence score of the AlphaFold2 structural prediction, specifically for the C-terminal portion of LptM, and we remove our emphasis on the possible interaction of LptM C-terminal region with loop 4 of LptD.

-Discussion, lines 587-588: We tone down our emphasis on the possible LptM-LptD loop 4 interaction and we specifically highlight the interaction of LptM with LptE in the lumen of the barrel.

The authors attempt to provide additional support for the AF2 model of LptM by site-specific photocrosslinking. While these data are reasonable for the N-terminal interaction with LptD, unfortunately, they are unable to observe any pXL between the C-term of LptM and LptE or LptD when the benzophenone is incorporated into LptM. This concern is potentially mitigated because they claim to observe crosslinks to LptM when the benzophenones are incorporated into LptE (Sup Fig 8e). But these data are not particularly convincing. First, the authors once again fail to validate the higher MW species by blotting for LptE. Could the complex banding around 27-32 kDa be something else? Moreover, crosslinks are claimed to be seen with LptE K70 but not K121 (which serves here as a negative control), despite the latter being closer to the AF2-modeled

position of LptM (and pointing towards it), and its location deep within the LptD barrel where it would presumably be less dynamic and more likely to crosslink.

In this paragraph the reviewer highlights that we have demonstrated an interaction of LptM N-terminal domain with LptD. Then the reviewer focuses on the lack of crosslinks between the LptM C-ter and the LPS translocon. As stated in our response point above (as well as in our manuscript) the LptM C-terminal region is unstructured and dynamic, which is likely to explain the lack of crosslink adducts when pBpa is incorporated in this part of LptM.

Then the reviewer requests additional controls about the positioning of LptM segments in the lumen of the LptD beta-barrel domain that we now provide.

In the first revised version of our manuscript we had illustrated an experiment showing that a region of LptE (K70, V80, I82 and A83) in the lumen of the LptD barrel can crosslink to LptM. Crosslinking efficiency was most efficient with pBpa at positions K70 and A83. V80, I82 and A83 are proximal to LptM, whereas LptE K70 appears more distal but points towards the periplasmic loop of LptM, which our MD simulations show to be dynamic and to lean towards LptE. In contrast, pBpa at position LptE K121 did not generate any detectable crosslink adduct. Thus, the experiment conducted with pBpa at position K121 served as control for the position-specificity of the crosslink reaction. LptE K121 is confined between the internal surface of the LptD beta-barrel and the LptD C-terminal segments in the lumen of the LptD forming a hydrogen bond with LptD N782 in our model. Such structural organization of LptE K121 would leave little room for an interaction with LptM, explaining the lack of a crosslink. We did not perform blotting with LptE-specific antibodies as LptE is the protein that contains pBpa and is thus the sole protein that can generate the UV-specific and site-specific crosslink adducts recognized by LptM antibodies.

To fully address the concern of the reviewer, we have now performed a new experiment (shown in the modified Supplementary Figure 8e) providing an improved demonstration that LptE interacts with LptM in the lumen of the LptD beta-barrel. We have repeated the crosslink experiment with pBpa engineered at positions K70 and A83 of LptE, which efficiently crosslink LptM. We have performed additional control reactions that rigorously demonstrate the identified LptE-LptM crosslink adducts depend on the incorporation of pBpa in LptE, as those do not form if pBpa is not added to the culture medium or if the wild-type lptE ORF (lacking any amber codon) is used. As suggested by the reviewer, we have now double checked that these crosslink adducts are immunodecorated not only with LptM-specific but also with LptE-specific antibodies. Our new experiment demonstrates without any ambiguity that LptE interacts with LptM in proximity of LptE positions K70 and A83, with the latter being positioned deep in the lumen of the LptD beta-barrel. This new experiment is shown in the modified supplementary Figure 8e (and described in the Results, lines 433-440), replacing the previous LptE-LptM crosslinking experiment.

Taken together, multiple approaches including native-MS (Figure 1c), structural prediction (Figures 5a-c), HDX-MS (Figure 5d,e), and the obtained crosslink adducts LptM-LptD and LptE-LptM (Supplementary Figures 8b-e) fully prove that LptM can form a ternary complex together with LptD and LptE. More specifically, our demonstrated association of LptM with LptD beta-barrel domain (Supplementary Figure 8a) and the crosslink of LptE K70 and A83 with LptM provide clear indication that parts of the small LptM protein are positioned in the lumen of LptD beta-barrel, consistent with our AlphaFold2 structural prediction.

The presentation of sequence conservation mapped to the structures of LptDEM (new Sup Fig 7) is confusing. Sequence conservation should be mapped to the structure of each component and then displayed in a manner that more clearly allows the reader to evaluate the proposed interaction surfaces.

Following the suggestion of the reviewer to tone down our arguments on the precise localization of LptM C-terminus, we have modified this figure to simply show the sequence conservation of LptM and LptD, and to highlight only the protein interfaces in proximity of the N-terminal segment of LptM, the positioning of which is predicted with a IDDT score > 50%.

Reviewers' Comments:

Reviewer #2:

Remarks to the Author:

The authors have addressed my concerns and I support publication of the work.